# ROBUST LEARNING FOR CONGESTION-AWARE ROUTING

## ABSTRACT

We consider the problem of routing users through a network with unknown congestion functions over an infinite time horizon. On each time step $t$, the algorithm receives a routing request and must select a valid path. For each edge $e$ in the selected path, the algorithm incurs a cost $c_e^t = f_e(x_e^t) + \eta_e^t$, where $x_e^t$ is the flow on edge $e$ at time $t$, $f_e$ is the congestion function, and $\eta_e^t$ is a noise sample drawn from an unknown distribution. The algorithm observes $c_e^t$, and can use this observation in future routing decisions. The routing requests are supplied adversarially.

We present an algorithm with cumulative regret $\tilde{O}(|E|t^{2/3})$, where the regret on each time step is defined as the difference between the total cost incurred by our chosen path and the minimum cost among all valid paths. Our algorithm has space complexity $O(|E|t^{1/3})$ and time complexity $O(|E| \log t)$. We also validate our algorithm empirically using graphs from New York City road networks.

## 1 INTRODUCTION

Modern navigation applications such as Google Maps and Apple Maps are critical tools in large scale mobility solutions that route billions of users from their source to their destination. In order to be effective, the application should be able to accurately estimate the time required to traverse each road segment (edge) along the user route in the road network (graph): we call this the *cost* of the edge. In general, the cost of an edge also depends on the current traffic level (the *flow*) on the edge. Furthermore, costs may scale differently on different edges: a highway can tolerate more traffic than a small residential street. We model this using *congestion functions* that map traffic flows to edge costs.

Usually, cost information is not readily available to the routing engine and can only be inferred indirectly. For example, this can be done using location pings from vehicles, or with loop detectors that record when vehicles cross a particular marker. All realistic methods of measuring the cost of an edge (such as those above) require the presence of a vehicle that reports this information back to the routing platform. In this paper, we assume that whenever a vehicle traverses an edge, we observe the time spent on the edge. We can then use this observation in future routing decisions. This induces a natural exploration/exploitation trade-off: we may wish to send vehicles on underexplored routes, even if those routes currently seem suboptimal.

In this paper, we propose a learning model for congestion-aware routing, and present an algorithm which seeks to minimize the total driving time across all vehicles. Our algorithm applies to arbitrary networks and arbitrary (Lipschitz-continuous) congestion functions, even when observations are noisy. The algorithm is also robust to changes in traffic conditions in a strong sense: we show that even when request endpoints are chosen adversarially and even when traffic congestion on the edges is updated adversarially between requests, our algorithm learns an optimal routing policy.

### 1.1 MODEL

Consider a directed graph $(V, E)$. Each edge $e$ has a deterministic and fixed (but unknown) congestion function $f_e : \mathbb{R}_{\geq 0} \to \mathbb{R}_{\geq 0}$. We assume each $f_e$ is $L$-Lipschitz continuous and nondecreasing, with $L$ known. For simplicity, we also assume that $f_e(0) = 0$ for all $e \in E$ (if not, we can simply translate and extend the function appropriately).

We consider an infinite horizon of time steps, starting from $t = 0$. At each time $t$, a new car arrives. An adversary tells us the current amount of flow on each edge, and the source and destination of the new car. Let $x_e^t$ be the flow on edge $e$ at time $t$ and let $P_t$ be the set of paths between the source and destination for the time $t$ arrival. Let $x_{max}^t = \max_{e,t} x_e^t$. We must choose how to route the car, i.e., we must select a path $p_t \in P_t$. Based on our choice of $p_t$, we incur a cost based on the flow and the congestion functions on the edges in our chosen path:

$$c_t = \sum_{e \in p_t} f_e(x_e^t)$$

For each $e \in p_t$, we observe $c_e^t = f_e(x_e^t) + \eta_e^t$, where $\eta_e^t$ is a random variable with expectation 0. The distribution of $\eta_e^t$ is unknown, and can vary between edges and time steps. The distributions can be correlated across edges, as long as for a given edge, all the individual samples (i.e., $\eta_e^1, \eta_e^2, \dots$) are independent. We assume that there exists $\beta$ such that $\eta_e^t \in [-\beta/2, \beta/2]$ for all edges $e$ and times $t$, and that $\beta$ (or an upper bound on $\beta$) is known.

The optimal cost at time $t$ is

$$c_t^* = \min_{p \in P_t} \sum_{e \in p} f_e(x_e^t)$$

so the *regret* of our algorithm over the first $t$ time steps is

$$R_t = \sum_{r=1}^{t} \mathbb{E}[c_t - c_t^*]$$

where the expectation is over the randomness in the noise samples. Note also that we do not include the noise we observe in the objective function, since the noise has expectation 0.

Any algorithm with sublinear regret can be said to learn an optimal routing policy: if $R_t = o(t)$, then $\mathbb{E}[c_t - c_t^*]$ must shrink as $t$ goes to infinity, i.e., the difference between our algorithm's cost and the optimal cost must go to 0 on average.[1] In this paper, we give an algorithm with regret $\tilde{O}(t^{2/3})$.

## 1.2 OUR CONTRIBUTION

Our main result is the following theorem:

**Theorem 1.1.** *The expected regret of Algorithm 1 after $t$ time steps is*

$$R_t = O\left(t^{2/3} \cdot |E| \log x_{max}^t (\beta \sqrt{\log t} + x_{max}^t L)\right)$$

*The space complexity and time complexity on time step $t$ are $O(|E| t^{1/3} \log x_{max}^t)$ and $O\Big(|E|(\log t + \log \log x_{max}^t) + SP(|E|, |V|)\Big)$, respectively.*

Here $SP(|E|, |V|)$ denotes the time complexity of computing the shortest path between two vertices in a graph with nonnegative weights. For example, this can be done by Dijkstra's algorithm in time $O(|E| + V \log V)$.

We also validate our algorithm's performance using graphs from New York City road networks.

## 1.3 RELATED WORK

**Comparison with multi-armed bandits.** Perhaps the simplest model of exploration vs exploitation is the classical *multi-armed bandit* (MAB) problem (Slivkins, 2019). A MAB instance consists of $n$ arms, each with an unknown but fixed distribution. At each time step, the algorithm selects an arm and observes a reward drawn randomly from that arm's distribution. Several algorithms obtaining regret $\tilde{O}(\sqrt{t})$ are known, and this is also known to be the best possible, up to logarithmic factors.

---

[1]Note that $\mathbb{E}[c_t - c_t^*]$ may not monotonically decrease: a sublinear regret algorithm can still incur large errors sporadically, as long as the the frequency of the large errors goes to 0.

Our routing model generalizes the MAB problem. In particular, when the graph consists of $n$ parallel edges, the congestion functions are constant, and each edge has a single fixed noise distribution, our model reduces to the MAB problem: each edge is an arm, and the reward distribution for each edge is (the negative of) the congestion constant plus the noise distribution. Our model can be thought of as extending the MAB problem to the case where (1) each reward distribution has a parameter (in our case, the flow), and (2) the arms are edges in an arbitrary graph and their usage is constrained according to paths in the graph. We believe that routing is the most natural application of this model, but there could be other applications as well.

Since our problem is strictly harder than the MAB problem, we know that regret $o(\sqrt{t})$ is impossible. We conjecture that even regret $\Theta(\sqrt{t})$ is impossible for our problem, but leave this as an open question.

**Comparison with other routing work.** Our work is also related to the body of literature on *shortest paths under uncertainty*, in which it is typically assumed that edge lengths follow known independent distributions. A canonical problem is to find an $s$–$t$ path whose length is below a threshold $L$ with highest probability. When the edge length distributions are Gaussian, Nikolova et al. (2006) present a quasi-polynomial time algorithm for this problem via connections to quasi-convex maximization. A related probing problem is the *Canadian Traveler Problem* (Nikolova & Karger, 2008; Papadimitriou & Yannakakis, 1991), where the length of an edge is revealed when we reach one of its end-points and the goal is to minimize the expected cost. There are no efficient algorithms known for this problem, except under special assumptions such as no backtracking (Bnaya et al., 2009).

Awerbuch & Kleinberg (2004) study a conceptually similar adaptive routing problem. In their version, there is no noise in the observations, but the algorithm observes only the total cost of the path, and not the cost of each individual edge. Their model also does not have the same notion of a parametrized congestion function. Although their model is technically distinct from ours, it is noteworthy that they also obtain a regret bound of $\tilde{O}(t^{2/3})$. A number of subsequent works have also sought to apply bandit algorithms to shortest path selection problems in the domain of network signal routing (Chen & Ji, 2005; György et al., 2007; Liu & Zhao, 2012; Zou et al., 2014; Talebi et al., 2018). Moreover, there is a long line of other works broadly dealing with regret minimization in Internet routing and congestion control settings (Dong et al., 2015; 2018; Jiang et al., 2016; 2017; Talebi et al., 2018).

Similar works apply bandit algorithms to shortest path selection have also been applied to traffic route planning (Chorus, 2010; de Oliveira Ramos et al., 2017). However, these works differ from ours in that they do not consider the more general congestion functions that we consider in our model. A more recent work (Zhou et al., 2019) considers a similar routing problem, labeled the Multi-Armed Bandit On-Time Arrival Problem. In this setting, regret is quantified in terms of on-time arrival reliability, and the arms are joint route and departure times. Kveton et al. (2014) also study (among other applications) a routing setting in the context of matroid bandits, but this work also assumes static edge costs and does not factor in congestion.

Routing in the presence of congestion functions has been studied in the context of selfish routing Roughgarden & Tardos (2002) and in the atomic congestion games setting Awerbuch et al. (2005); Christodoulou & Koutsoupias (2005). The difference from our setting is that the congestion games model and selfish routing models assume drivers have full information about the costs in the network and route themselves on an optimal path. In our setting costs are learned as time progresses and there is a centralized routing control as given by navigation applications.

## 2 DESCRIPTION OF THE ALGORITHM

In this section, we define and give intuition for our algorithm for learning an optimal routing policy.

### 2.1 INTUITION BEHIND THE ALGORITHM

On each step, the algorithm uses past observations to form an estimate for the cost of each edge at the current flow, i.e., an estimate for $f_e(x_e^t)$. It then selects the shortest path according to these estimated costs.

The heart of the algorithm is the cost estimation scheme. There are two sources of error in this estimation: we may not have observed this exact flow before (type 1 error), and noise (type 2 error). For type 1 error, suppose all the observations we use are based on flows $y$ such that $|y - x_e^t| \leq \varepsilon$. Then Lipschitz continuity implies that $|f_e(y) - f_e(x_e^t)| \leq \varepsilon L$, and so if $\varepsilon$ is small, this aspect of our estimation should be accurate.

However, we do not actually observe $f_e(y)$: instead, we observe $f_e(y) + \eta_y$, where $\eta_y$ is a random noise term with expectation 0. Even if we have observed the exact flow $x_e^t$ before, this noise will prevent us from having a perfect estimate. However, the more observations we use to form our estimate, the less impact the noise has: we can use Hoeffding's Inequality to show that $|\sum_y \eta_y|$ goes to 0 quickly as the number of observations grows, and thus our type 2 error shrinks. But as we increase the number of observed flows $y$ that we use (for a fixed time step), we also (in general) increase the maximum distance $\max_y |y - x_e^t|$. Concretely, if $k(\varepsilon, x)$ is the number of observations based on flows within $\varepsilon$ of $x$, then $k(\varepsilon, x)$ is weakly increasing with $\varepsilon$. This is diametrically opposed to our plan for the type 1 error.

In order for the average error to approach 0 as $t$ increases, we will have to carefully manage this tradeoff. We will need to ensure that the number of observations we use for estimation tends to infinity, but also that the maximum distance tends to 0. Intuitively, this should be possible: if observations are somewhat uniformly distributed across edges and across the flow spectrum, then we should have $\Theta(t)$ observations per edge and $\Theta(t\varepsilon/x_{max}^t)$ observations on every interval of size $\varepsilon$ (treating $|E|$ as a constant here). Setting $\varepsilon$ to be something like $\Theta(t^{-1/2})$ would lead to $\Theta(\sqrt{t}/x_{max}^t)$ observations being used for each estimation with a maximum distance of $\Theta(t^{-1/2})$, which fits our requirements.

Observations are of course not guaranteed to be uniform, and can even be adversarial. For example, suppose we have observed many flows less than 1 on some edge $e$, but no flows above 1. When estimating the cost of a flow less than 1, we are in good shape (if those observations are uniformly distributed, say). But suppose we are asked to estimate the cost for flow 2: we may incur a large estimation error, since we have no information about that portion of the flow spectrum.

But our lack of information also implies that so far we have not incurred any error in that part of the flow spectrum, since we have never used it! The more we use part of the flow spectrum on a given edge (i.e., our chosen path used that edge at a flow in that range), the more error we incur, but the more we learn. Thus we will need an analysis that is decomposable across edges and across the flow spectrum: for each interval $I$, we need to bound our cumulative error over all estimation queries that fall within that interval as a function of the length of the interval and the number of such queries.

There is one more complicating factor: what if we consistently overestimate the cost of a particular flow, so that we never use it? Then we may repeatedly incur a large error from that flow, without ever learning about it? For this reason, we will want to ensure that with high likelihood, all of our estimates are underestimates: this will allow us to bound the regret on a given time step in terms of the estimation error *only on edges in the path that we used*.

## 2.2 OUR COST ESTIMATION SCHEME

The cost estimation works roughly as follows. The algorithm maintains a "bucketing" system for each edge, inspired by hashtables. The interval $[0, x_{e,max}^t]$ is partitioned into a set of intervals, with each interval being a "bucket". Whenever we observe a new flow, we identify which bucket that flow falls into, and use all the observations in that bucket to estimate the cost. (We also subtract a term proportional to $\sqrt{\log t}$ in order to ensure that we get an underestimate with high probability; this is similar to the Upper Confidence Bound algorithm for multi-armed bandits.) We then observe the resulting cost for this flow (for this edge), and insert it into the bucket. If we observe a flow that falls outside of our bucketing system, we create a new depth 0 bucket so that the observed flow falls into our new bucket.

If the number of elements in the bucket surpasses a certain threshold, we split it into two new buckets (whose associated intervals are equal length). The threshold depends on the depth (i.e., how many times one of its ancestors was split). We denote the "lifetime" of a bucket at depth $m$ by and we denote it by $h(m)$. Our analysis will be "decomposable" across buckets: we will bound the total error any single bucket of depth $m$ can contribute over the course of its lifetime.

Let $b_e^t$ be the bucket we used for estimating the cost of edge $e$ at time $t$. The smaller the interval associated with this $b_e^t$, the smaller the error of type 1: if our estimation only uses observed flows $y$ within this interval, and the interval has length $\varepsilon$, then the type 1 error is at most $\varepsilon L$, since our target flow $x_e^t$ is also in the interval. This incentivizes us to make $h(m)$ small, so that we can more quickly we get to buckets with small interval lengths. If we only cared about type 1 error (i.e., if there were no noise) choosing $h(m) = 1$ would be optimal.

On the other hand, the more observations stored in $b_e^t$, the smaller type 2 error. This incentivizes us to make $h(m)$ large, so that each bucket contains a large number of observations for a larger portion of its lifetime. If we only cared about type 2 error (i.e., if $f_e$ were constant), choosing $h(m) = \infty$ would be optimal. This is exactly the tradeoff we discussed in Section 2.1. Eventually, $h(m) = 2^{2m}$ will give us the best regret bound.

### 2.3    DEFINING THE ALGORITHM

Algorithm 1 provides pseudocode for our algorithm. We use $B_e$ to denote the entire bucketing system for edge $e$. We treat $B_e$ as a set, whose elements are "buckets". Each bucket $b$ supports the following operations:

$$|b| \rightarrow \text{returns the number of (flow, cost) pairs stored in } b$$
$$\mathrm{dom}(b) \rightarrow \text{returns the interval } [w, z] \subseteq [0, K] \text{ associated with } b$$
$$\mathrm{depth}(b) \rightarrow \text{returns the depth of bucket } b$$
$$\mathtt{estimate}(b) \rightarrow \text{returns the cost estimate based on the current observations stored in } b$$
$$\mathtt{create}(m, [w, z]) \rightarrow \text{returns a new empty bucket } b \text{ with } \mathrm{dom}(b) = [w, z] \text{ and } \mathrm{depth}(b) = m$$
$$\text{INSERT}(b, (y, c_y)) \rightarrow \text{inserts the flow } y \text{ and its associated observed cost } c_y \text{ into } b$$

The first four operations will simply be numerical fields stored in $b$. We also assume that the $\mathtt{create}$ operation takes constant time and space, and that the aforementioned four fields are initialized appropriately ($\mathtt{estimate}(b)$ should be 0). However, we will need to implement INSERT ourselves in order to ensure that it is done in an efficient fashion (hence the differing font). In general, we will reserve the variables $w, z$ for $\mathrm{dom}(b)$, $y$ for arbitrary elements of $b$, and $x = x_e^t$ for the proposed flow on the current time step.

To limit space usage, we will not actually "store" each relevant observation in the bucket; instead, we will simply incorporate it into the $\mathtt{estimate}$ field that we store. However, in the analysis, it will be useful to use "$y \in b$" to denote that a particular observation $(y, c_y)$ has been used in that bucket. For each $y \in b$, let $\eta_y$ denote the noise sample associated with this observation. Note that $\eta_y$ is not known to the algorithm; we will simply use this notation in our analysis. Also, let $\mathtt{len}(\mathrm{dom}(b))$ denote the length of the domain, i.e., $z - w$ if $\mathrm{dom}(b) = [w, z]$.

We can think of the bucketing system in terms of binary trees. Each bucket of depth 0 is a "root", and whenever we split a bucket, we create two "children". Thus $B_e$ is essentially a forest of binary trees, where each node represents a bucket. Any bucket $b \in B_e$ has exactly one ancestor of depth 0.

## 3    EXPERIMENTS

In this section, we empirically validate the algorithm's performance on graphs from New York City road networks. We first describe the methodology, and then reflect on the results.

### 3.1    METHODOLOGY

We used two different graphs, corresponding to different regions within New York City. Graph 1 consisted of 303 vertices and 657 edges, and Graph 2 consisted of 429 vertices and 1204 edges.Congestion functions with domain $[0, 1]$ were generated randomly for each run of the algorithm, with each congestion function consisting of three pieces of slope chosen uniformly from $[0, 1]$. Uniform noise distributions were used for all runs, with varying values of $\beta$. The algorithm was initialized with $L = 1$ and $\alpha = 2\beta^2$. On each time step, the following steps were performed:

1. Independently sample a uniformly random flow $x_e^t$ in $[0, 1]$ for each edge $e$.

---

**Algorithm 1** Algorithm for learning an optimal social routing policy.

1: **function** ROUTINGALGORITHM($E, L, \alpha, h$)
2:     **for each** $e \in E$ **do**
3:         $B_e \leftarrow \{\texttt{create}(0, [0, 1])\}$   ▷ Start with a single bucket of depth 0 and constant length
4:     **for each** $t \in \mathbb{N}_{>0}$ **do**
5:         $x_e^t \leftarrow$ proposed flow on edge $e$
6:         **for each** $e \in E$ **do**
7:             $u_e^t \leftarrow$ ESTIMATEEDGECOST($x_e^t, B_e, L, \alpha$)
8:         $p_t \leftarrow \arg\min_{p \in P_t} \sum_{e \in p} u_e^t$
9:         $\texttt{selectRoute}(p_t)$, observe cost $c_e^t \; \forall e \in p_t$
10:        **for each** $e \in p_t$ **do**
11:            UPDATEBUCKETS($x_e^t, B_e, c_e^t, h$)

1: **function** ESTIMATEEDGECOST($x, B, L, \alpha, t$)
2:     **if** $\exists b \in B$ s.t. $x \in \texttt{dom}(b)$ and $|b| > 0$ **then**
3:         **return** $\max\left(0, \texttt{estimate}(b) - \sqrt{|b|^{-1} \ln t^\alpha}\right)$
4:     **else**
5:         **return** 0

1: **function** UPDATEBUCKETS($y, c_y, B, h$)
2:     **if** $\exists b \in B$ s.t. $y \in \texttt{dom}(b)$ **then**       ▷ Insert our observation into the corresponding bucket
3:         INSERT($b, (y, c_y)$)
4:         $m \leftarrow \texttt{depth}(b)$
5:         **if** $|b| > h(m)$ **then**                  ▷ Split the bucket if it is too big
6:            $[w, z] \leftarrow \texttt{dom}(b)$
7:            $b_1 \leftarrow \texttt{create}(m + 1, [w, \frac{w+z}{2}])$
8:            $b_2 \leftarrow \texttt{create}(m + 1, [\frac{w+z}{2}, z])$
9:            INSERT($b_1, (y, c_y)$)  ▷ Insert the observation into new buckets so they aren't empty.
10:           INSERT($b_2, (y, c_y)$)       ▷ Type 1 error is still proportional to the domain lengths.
11:           $B_e \leftarrow (B_e \setminus b) \cup \{b_1\} \cup \{b_2\}$
12:     **else**                  ▷ This flow does not fall into any bucket: create a new depth 0 bucket
13:         $y_{max} \leftarrow \max\left(\cup_{b \in B_e} \texttt{dom}(b)\right)$
14:         $b \leftarrow \texttt{create}\left(0, [y_{max}, 2x]\right)$
15:         $B_e \leftarrow B_e \cup \{b\}$

1: **function** INSERT($b, (y, c_y)$)
2:     $[w, z] \leftarrow \texttt{dom}(b)$
3:     **if** $y < w$ **then**                ▷ $y \notin \texttt{dom}(b)$ is possible if we created $b$ on this time step
4:         $u_y = c_y$
5:     **else**
6:         $u_y = c_y - L(y - w)$
7:     $\texttt{estimate}(b) \leftarrow \dfrac{|b| \cdot \texttt{estimate}(b) + u_y}{|b| + 1}$    ▷ This is a running average of $\{u_y : y \in b\}$
8:     $|b| \leftarrow |b| + 1$

---

2. Independently sample a uniformly random noise term $\eta_e^t \in [-\beta/2, \beta/2]$ for each edge $e$.

3. Independently sample a uniformly random source and destination from the graph.

4. Compute the path chosen $p_t$ chosen by our algorithm.

5. Compute $c_t - c_t^* = \sum_{e \in p_t} f_e(x_e^t) - \min_{p \in P_t} \sum_{e \in p} f_e(x_e^t)$ and record the cumulative regret $R_t$.

Each run was characterized by the following parameters:

1. Time horizon (either 100,000 or 1,000,000)

2. The graph used (Graph 1 or Graph 2)

3. The noise parameter $\beta \in \{0, .01, .02, .05, .1, .2, .5\}$.[2]

Figure 1 shows the results of all runs.

## 3.2 RESULTS

All runs of the algorithm display the general behavior we would hope to see: concavity of the cumulative regret curve. Also as expected, higher noise levels lead to both larger absolute regret, and weaker concavity: learning takes longer. Most of the runs display substantial concavity within 100,000 time steps, but some the $\beta = .5$ runs (on both graphs) require closer to 200,000 steps in order for the concavity to clearly manifest.

With regards to absolute regret, $\beta$ has a dramatic effect. On the extremes, the total regret for $\beta = 0$ after 1,000,000 time steps is less than 5,000 for Graph 1 and less than 9,000 for Graph 2. (The regret for Graph 1 tends to larger, due to the graph being larger.) However, for $\beta = .5$, the regret after 100,000 time steps is already more than 12,000 and 20,000 for Graph 1 and Graph 2, respectively. If we look at the average regret for $\beta = 0$ over 1,000,000 time steps, we get $<.005$ and $<.009$ for Graphs 1 and 2, respectively. Since the expected cost of each edge on a given time step is .25 (expected flow is .5, expected slope is .5), we view this as quite good. In contrast, for $\beta = .5$, the average regret over 1,000,000 time steps is $> .48$ and $>.88$ for Graphs 1 and 2 respectively. Relative to the average edge cost of .25, we view this as quite poor. This behavior again matches our expectations.

Finally, we note that there are many other experimental parameters that we have not included in these plots. For example: non-i.i.d flows, non-i.i.d. sources and destinations, non-i.i.d. congestion functions, congestion functions with different numbers of pieces, congestion functions with different Lipschitz constants, other noise distributions, correlated noise, and many more. We have indeed tested many of these, but due to space constraints, there is a limit to the number of plots we can include in the paper. We assure the reader that we have yet to find a parameter combination that causes unexpected behavior from the algorithm.

## 4 CONCLUSION

In this paper, we presented an algorithm which learns an optimal routing policy for any graph, any (Lipschitz continuous) congestion functions, and any (bounded) noise distribution, and adversarial routing requests (i.e., source, destination, and the current flow on each edge). Our algorithm has cumulative regret $\tilde{O}(|E|t^{2/3})$.

There are many interesting directions for future work. The first relates to the dependence on $t$ in the regret bound. Since our problem generalizes the multi-armed bandit problem, we immediately inherit a $\Omega(\sqrt{t})$ lower bound. However, that still leaves substantial room to improve the dependence on $t$. It would also be interesting to improve the dependence on $|E|$. We have made several worst-case assumptions in this paper that are perhaps overly pessimistic: for example, neighboring edges may have correlated congestion functions, and flows do not change arbitrarily between time steps.

On the empirical side, we evaluated our algorithm using real-world graphs, but synthetic congestion functions. Future experiments could use real data for the congestion functions as well: either by fitting a congestion function to the data, or by randomly sampling data points to simulate a congestion function.

---

[2]With regards to the chosen range of $\beta$ values, note that the expected maximum cost on each edge is .5, since the domain of each $f_e$ is $[0, 1]$ and the expected slope of each piece is .5. Thus in some sense, $\beta = .5$ represents "as much noise as signal", $\beta = .2$ represents "2/5 as much noise as signal", and so on.

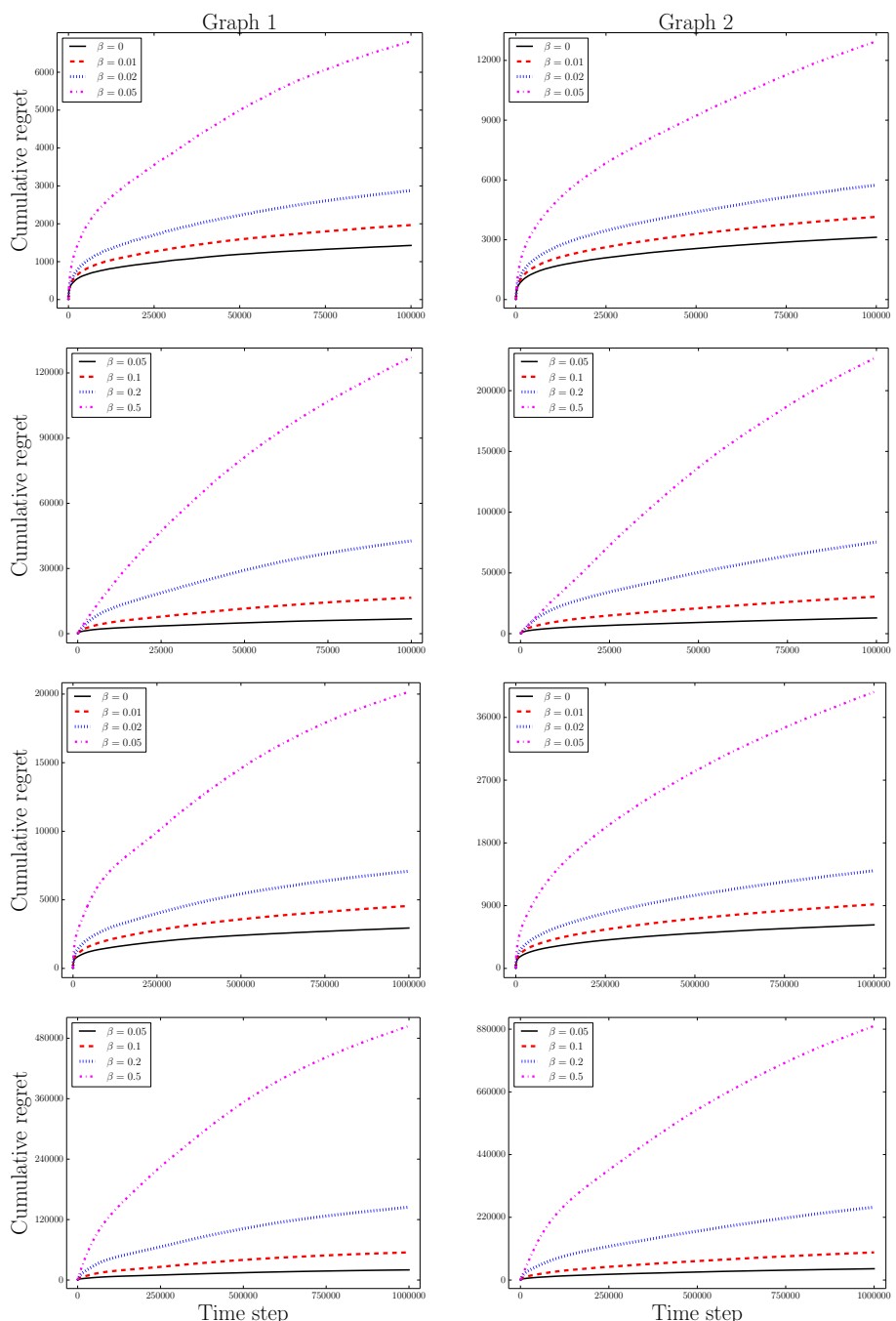

Figure 1: the experimental performance of Algorithm 1. The x-axis in each plots is the time step, and the y-axis is the cumulative regret. There are eight plots, each consisting of four curves, each with different noise levels. Each curve is a single run of the algorithm. The upper four plots have time horizons of 100,000, and the bottom four have time horizons of 1,000,000. All plots in the left column use Graph 1, and all plots in the right column use Graph 2. Finally, since we tested seven values of $\beta$, we separated them into two different plots to avoid overcrowding a single plot. For example, the top two plots in the left column have the same time horizon of 100,000 and both use Graph 1. The top left plot contains the lower noise levels (0, .01, .02, .05), and the second-from-the-top left contains the higher noise levels (.05, .1, .2, .5). Note that $\beta = .05$ is included in both to allow visual comparison between the plots.

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

## A   PROOF OF THEOREM 1.1

In this section, we prove our main theorem:

**Theorem 1.1.** *The expected regret of Algorithm 1 after $t$ time steps is*

$$R_t = O\left(t^{2/3} \cdot |E| \log x_{max}^t (\beta \sqrt{\log t} + x_{max}^t L)\right)$$

*The space complexity and time complexity on time step $t$ are $O(|E|t^{1/3} \log x_{max}^t)$ and $O\left(|E|(\log t + \log\log x_{max}^t) + SP(|E|, |V|)\right)$, respectively.*

We restate the algorithm below for the reader's convenience.

Our first lemma upper bounds the distance between two flows associated with the same bucket.

**Lemma A.1.** *Consider an arbitrary edge $e$ and bucket $b \in B_e$, let $(x, c_x) \in b$, and let $y \in dom(b)$ (but not necessarily $y \in b$). If $m = \mathtt{depth}(b)$, then $|x - y| \leq x_{max}^t / 2^{m-1}$.*

*Proof.* Let $b'$ be the unique depth 0 ancestor of $b$. If $b$ has depth $m$, then $\mathtt{dom}(b)$ must have length exactly $\mathtt{len}(\mathtt{dom}(b'))/2^m$, since the domain length is halved whenever the depth is incremented. Since $\mathtt{len}(\mathtt{dom}(b')) \leq x_{max}^t$, we have $\mathtt{len}(\mathtt{dom}(b)) \leq x_{max}^t/2^m$.

If $x \in \mathtt{dom}(b)$, then $|x - y| \leq x_{max}^t/2^m$ and we are done. If not, then $x$ must have been inserted on the time step that $b$ was created. Let $b''$ be the parent of $b$. In this case, $x \in \mathtt{dom}(b'')$. By construction, $\mathtt{dom}(b) \subset \mathtt{dom}(b')$, so we have $x, y \in \mathtt{dom}(b'')$. Finally, since $b'$ has depth $m - 1$, $\mathtt{len}(\mathtt{dom}(b'')) \leq x_{max}^t/2^{m-1}$, so $|x - y| \leq x_{max}^t/2^{m-1}$.   □

Lemma A.2 is a standard concentration equality due to Hoeffding. Lemma A.3 is essentially a restatement of Hoeffding's Lemma for our setting.

---

**Algorithm 1** Algorithm for learning an optimal social routing policy.

1: **function** ROUTINGALGORITHM($E, L, \alpha, h$)
2:    **for each** $e \in E$ **do**
3:        $B_e \leftarrow \{\texttt{create}(0, [0, 1])\}$   ▷ Start with a single bucket of depth 0 and constant length
4:    **for each** $t \in \mathbb{N}_{>0}$ **do**
5:        $x_e^t \leftarrow$ proposed flow on edge $e$
6:        **for each** $e \in E$ **do**
7:            $u_e^t \leftarrow$ ESTIMATEEDGECOST($x_e^t, B_e, L, \alpha$)
8:        $p_t \leftarrow \arg\min_{p \in P_t} \sum_{e \in p} u_e^t$
9:        $\texttt{selectRoute}(p_t)$, observe cost $c_e^t \; \forall e \in p_t$
10:       **for each** $e \in p_t$ **do**
11:           UPDATEBUCKETS($x_e^t, B_e, c_e^t, h$)

1: **function** ESTIMATEEDGECOST($x, B, L, \alpha, t$)
2:    **if** $\exists b \in B$ s.t. $x \in \texttt{dom}(b)$ and $|b| > 0$ **then**
3:        **return** $\max\left(0, \texttt{estimate}(b) - \sqrt{|b|^{-1} \ln t^\alpha}\right)$
4:    **else**
5:        **return** $0$

1: **function** UPDATEBUCKETS($y, c_y, B, h$)
2:    **if** $\exists b \in B$ s.t. $y \in \texttt{dom}(b)$ **then**        ▷ Insert our observation into the corresponding bucket
3:        INSERT($b, (y, c_y)$)
4:        $m \leftarrow \texttt{depth}(b)$
5:        **if** $|b| > h(m)$ **then**                                    ▷ Split the bucket if it is too big
6:            $[w, z] \leftarrow \texttt{dom}(b)$
7:            $b_1 \leftarrow \texttt{create}(m + 1, [w, \frac{w+z}{2}])$
8:            $b_2 \leftarrow \texttt{create}(m + 1, [\frac{w+z}{2}, z])$
9:            INSERT($b_1, (y, c_y)$)   ▷ Insert the observation into new buckets so they aren't empty.
10:           INSERT($b_2, (y, c_y)$)            ▷ Type 1 error is still proportional to the domain lengths.
11:           $B_e \leftarrow (B_e \setminus b) \cup \{b_1\} \cup \{b_2\}$
12:   **else**                              ▷ This flow does not fall into any bucket: create a new depth 0 bucket
13:       $y_{max} \leftarrow \max\left(\cup_{b \in B_e} \texttt{dom}(b)\right)$
14:       $b \leftarrow \texttt{create}(0, [y_{max}, 2x])$
15:       $B_e \leftarrow B_e \cup \{b\}$

1: **function** INSERT($b, (y, c_y)$)
2:    $[w, z] \leftarrow \texttt{dom}(b)$
3:    **if** $y < w$ **then**                        ▷ $y \notin \texttt{dom}(b)$ is possible if we created $b$ on this time step
4:        $u_y = c_y$
5:    **else**
6:        $u_y = c_y - L(y - w)$
7:    $\texttt{estimate}(b) \leftarrow \dfrac{|b| \cdot \texttt{estimate}(b) + u_y}{|b| + 1}$       ▷ This is a running average of $\{u_y : y \in b\}$
8:    $|b| \leftarrow |b| + 1$

---

**Lemma A.2** (Hoeffding's Inequality). *Let $X_1, \ldots, X_n$ be independent random variables each supported on an interval of length $\beta > 0$ and let $\bar{X} = \frac{1}{n} \sum_{i=1}^{n} X_i$. Then for any $\varepsilon > 0$,*

$$Pr[|\bar{X} - \mathbb{E}[\bar{X}]| > \varepsilon] \leq \exp\left(\frac{-2n\varepsilon^2}{\beta^2}\right)$$

**Lemma A.3.** *Fix an edge $e$ and a bucket $b \in B_e$. With probability at least $1 - \delta$,*

$$\left|\frac{1}{|b|} \sum_{y \in b} \eta_y\right| \leq \beta \sqrt{\frac{\ln \delta^{-1}}{2|b|}}$$

*Proof.* If $\beta = 0$, the claim is trivially true. Otherwise, since $\mathbb{E}[\eta_y] = 0$, and all $\eta_y$ are independent for a given edge, Hoeffding's Inequality gives us

$$\Pr\left[\left|\frac{1}{|b|}\sum_{y \in b}\eta_y\right| > \beta\sqrt{\frac{\ln \delta^{-1}}{2|b|}}\right] \leq \exp\left(\frac{\beta^2 \ln \delta^{-1}}{2|b|} \cdot \frac{-2|b|}{\beta^2}\right) = \delta$$

as required. $\qquad\square$

We will now start the process of bounding the costs incurred by our algorithm. Moving forward, we will need to separately handle the time steps where $x_e^t$ does not fall into any bucket we have, and we must create a new depth 0 bucket (on line 14 of UPDATEBUCKETS). Let $T_e$ be the set of time steps $t$ where (1) $e \in p_t$, (2), $\exists r < t$ such that $e \in p_r$ (i.e., this is not the first we are using this edge), and (3) we do *not* create a new depth 0 bucket. The reader can think of $T_e$ as the set of time steps when edge $e$ is used in a "normal" way.

Throughout the analysis, for each $e \in E$ and time $t$, let $b_e^t$ denote the bucket that is used on line 3 of ESTIMATEEDGECOST, if such a bucket exists. For each $t \in T_e$, we are guaranteed that $b_e^t$ exists, since Condition 3 above ensures that there exists $b \in B_e$ with $x_e^t \in \text{dom}(b)$, and Condition 2 ensures that $|b| > 0$. Throughout the proof, we will use $|b_e^t|$ to denote the number of observations in the bucket *on that time step*, before inserting the new element (if one is inserted).

Also, for each $e \in E$ and time $t$, let $\zeta_e(t)$ be the indicator variable that $|\frac{1}{|b_e^t|}\sum_{y \in b_e^t}\eta_y| \leq \sqrt{|b_e^t|^{-1}\ln t^{\alpha}} = \beta\sqrt{|b_e^t|^{-1}\ln t^{\alpha/\beta^2}}$. (If $b_e^t$ does not exist, let $\zeta_e(t) = 1$). Note that by Lemma A.3, $\Pr[\zeta_e(t) = 1] \geq 1 - t^{-\alpha/\beta^2}$.

The next lemma states that as long as $\zeta_e(t) = 1$, our algorithm's estimate $u_e^t$ will always be an *underestimate* of the true cost $f_e(x_e^t)$.

**Lemma A.4.** *Fix a time $t$ and edge $e$. If $\zeta_e(t) = 1$, then $f_e(x_e^t) \geq u_e^t$.*

*Proof.* If $u_e^t = 0$, then trivially $u_e^t \leq f_e(x_e^t)$. Thus assume $u_e^t > 0$. This implies $t \in T_e$, and so $b_e^t$ exists. For each $y \in b_e^t$, define $u_y$ as in the INSERT function in Algorithm 1, and let $[w, z] = \text{dom}(b_e^t)$. For brevity, let $x = x_e^t$. Note that $x \geq w$ and thus $f_e(x) \geq f_e(w)$. Also recall that $c_y = f_e(y) + \eta_y$ by definition. We first claim that $f_e(x) \geq u_y - \eta_y$ for all $y \in b_e^t$.

Case 1: $w > y$. Then $x > y$, so we have $u_y - \eta_y = c_y - \eta_y = f_e(y) \leq f_e(x)$, as required.

Case 2: $y \geq w$. Then $u_y - \eta_y = c_y - \eta_y - L(y-w) = f_e(y) - L(y-w)$. Because $f_e$ is $L$-Lipschitz, we have $f_e(y) \leq f_e(w) + L(y-w)$, so

$$\begin{aligned} u_y - \eta_y &= f_e(y) - L(y-w) \\ &\leq f_e(w) + L(y-w) - L(y-w) \\ &\leq f_e(x) \end{aligned}$$

Therefore

$$\begin{aligned} f_e(x) &\geq \frac{1}{|b_e^t|}\sum_{y \in b_e^t}(u_y - \eta_y) \\ &= \frac{1}{|b_e^t|}\sum_{y \in b_e^t}u_y - \frac{1}{|b_e^t|}\sum_{y \in b_e^t}\eta_y \\ &\geq \frac{1}{|b_e^t|}\sum_{y \in b_e^t}u_y - \frac{1}{|b_e^t|}\left|\sum_{y \in b_e^t}\eta_y\right| \end{aligned}$$

Since $\zeta_e(t) = 1$ by assumption, we get

$$f_e(x_e^t) \geq \frac{1}{|b_e^t|}\sum_{y \in b_e^t}u_y - \sqrt{|b_e^t|^{-1}\ln t^{\alpha}} = \texttt{estimate}(b_e^t) - \sqrt{|b_e^t|^{-1}\ln t^{\alpha}}$$

Since $u_e^t > 0$, we have $u_e^t = \texttt{estimate}(b_e^t) - \sqrt{|b_e^t|^{-1}\ln t^{\alpha}}$. Thus $f_e(x_e^t) \geq u_e^t$, as required. $\quad\square$

Lemma A.5 states that if $t \in T_e$ and $\zeta_e(t) = 1$, then our estimate $u_e^t$ also shouldn't be too much more than $f_e(x_e^t)$.

**Lemma A.5.** *Fix a time $t \in T_e$ and edge $e$, and let $m = \texttt{depth}(b_e^t)$. If $\zeta_e(t) = 1$, then*

$$f_e(x_e^t) \leq u_e^t + \frac{x_{max}^t L}{2^{m-1}} + 2\sqrt{|b_e^t|^{-1} \ln t^{\alpha}}$$

*Proof.* As before, for each $y \in b_e^t$, define $u_y$ as in INSERT, let $[w, z] = \texttt{dom}(b_e^t)$, and let $x = x_e^t$ for brevity. We first claim that $f_e(x) \leq u_y - \eta_y + L(x - y)$ for all $y \in b_e^t$. Since $x \geq w$ and $f_e$ is $L$-Lipschitz, we have $f_e(x) \leq f_e(w) + L(x - w)$.

Case 1: $y \geq w$. Then $f_e(w) \leq f_e(y)$ by monotonicity, so

$$
\begin{aligned}
f_e(x) &\leq f_e(y) + L(x - w) \\
&= c_y - \eta_y + L(x - y) + L(y - w) \\
&= u_y - \eta_y + L(x - w) \\
&\leq u_y - \eta_y + 2^{1-m} x_{max}^t L
\end{aligned}
$$

where the last step follows from Lemma A.1.

Case 2: $w > y$.[3] Then $f_e(w) \leq f_e(y) + L(w - y)$, so

$$
\begin{aligned}
f_e(x) &\leq f_e(y) + L(w - y) + L(x - w) \\
&= c_y - \eta_y + L(x - y) \\
&\leq u_y - \eta_y + L(x - y) \\
&\leq u_y - \eta_y + 2^{1-m} x_{max}^t L
\end{aligned}
$$

again using Lemma A.1 on the last step. Therefore

$$
\begin{aligned}
f_e(x) &\leq \frac{1}{|b_e^t|} \sum_{y \in b_e^t} (u_y - \eta_y + 2^{1-m} x_{max}^t L) \\
&= 2^{1-m} x_{max}^t L + \frac{1}{|b_e^t|} \sum_{y \in b_e^t} u_y - \frac{1}{|b_e^t|} \sum_{y \in b_e^t} \eta_y \\
&\leq 2^{1-m} x_{max}^t L + \texttt{estimate}(b_e^t) + \frac{1}{|b_e^t|} \left| \sum_{y \in b_e^t} \eta_y \right| + \sqrt{|b_e^t|^{-1} \ln t^{\alpha}} - \sqrt{|b_e^t|^{-1} \ln t^{\alpha}}
\end{aligned}
$$

Since $u_e^t = \max(0, \texttt{estimate}(b) - \sqrt{|b|^{-1} \ln t^{\alpha}})$, we have $\texttt{estimate}(b) - \sqrt{|b|^{-1} \ln t^{\alpha}} \leq u_e^t$. Also, $\left| \frac{1}{|b_e^t|} \sum_{y \in b_e^t} \eta_y \right| \leq \sqrt{|b_e^t|^{-1} \ln t^{\alpha}}$, since $\zeta_e(t) = 1$ by assumption. Therefore

$$
\begin{aligned}
f_e(x) &\leq u_e^t + 2^{1-m} x_{max}^t L + \frac{1}{|b_e^t|} \left| \sum_{y \in b_e^t} \eta_y \right| + \sqrt{|b_e^t|^{-1} \ln t^{\alpha}} \\
&\leq u_e^t + 2^{1-m} x_{max}^t L + 2\sqrt{|b_e^t|^{-1} \ln t^{\alpha}}
\end{aligned}
$$

$\square$

The next lemma is a simple bound on the maximum flows on time steps when we create a new depth 0 bucket. This will be needed later to bound the error on those time steps.

**Lemma A.6.** *Fix an edge $e \in E$ and a time step $t > 0$. Let $t_0, \ldots, t_k$ be the time steps up to and including time $t$ on which a new depth 0 bucket is created in $B_e$ (where $t_0 = 0$). Then $k \leq 1 + \log x_{max}^t$. Furthermore, $\sum_{i=1}^{k} x_e^{t_k} \leq 2 x_{max}^t$.*

---

[3]This means that $y$ is actually not in $\texttt{dom}(b_e^t)$. This only happens if $y$ was inserted into $b_e^t$ on the time step when $b_e^t$ was created.

*Proof.* Assume $t_0 < t_1, \ldots, < t_k$. Let $y_{max}^r = \max(\cup_{b \in B_e^r} \mathrm{dom}(b))$, where $B_e^r$ denotes the set of buckets for edge $e$ at the beginning of time step $r$. This is the maximum flow that falls into any bucket in $B_e$ at time $r$ Note that $y_{max}^{r+1} = y_{max}^r$ for all $r \notin \{t_1, \ldots, t_k\}$. Furthermore, for $r \in \{t_1, \ldots, t_k\}$, we have $y_{max}^{r+1} = 2x_e^r > y_{max}^r$. Therefore for each $i \in \{1, \ldots, k\}$,

$$y_{max}^{t_{i+1}} = 2x_e^{t_i} > 2y_{max}^{t_i}$$

Thus for all $i, j \in \{1, k\}$ with $i \leq j$, $y_{max}^{t_j} > 2^{j-i}$. In particular,

$$y_{max}^{t_k} > 2^k \cdot y_{max}^{t_0} = 2^i$$

where $y_{max}^{t_0} = 1$ is because we initialize $B_e$ with a single bucket corresponding to the interval $[0, 1]$. We know that $y_{max}^{t_k+1} = 2x_e^{t_k-1} \leq 2x_{max}^t$, so $2^k \leq 2x_{max}^t$. Therefore $|D_e^t| = k \leq 1 + \log x_{max}^t$. Finally,

$$\sum_{i=1}^{k} x_e^{t_i} = \frac{1}{2} \sum_{i=1}^{k} y_{max}^{t_i+1}$$

$$\leq \frac{y_{max}^{t_k+1}}{2} \sum_{i=1}^{k} \frac{1}{2^{k-i}}$$

$$\leq \frac{y_{max}^{t_k+1}}{2} \cdot 2$$

$$\leq 2x_{max}^t$$

as required. $\qquad\square$

Lemma A.7 gives our first bound on the cumulative regret of the algorithm. This bound is quite complex, and it will take substantial work later on to show that this bound is in fact $\tilde{O}(t^{2/3})$.

**Lemma A.7.** *Assume $\alpha = 2\beta^2$. Then for all t, we have*

$$R_t \leq \frac{|E|x_{max}^t L(\pi^2 + 18)}{6} + \sum_{r=1}^{t} \sum_{e \in p_r} \left( \frac{x_{max}^t L}{2^{depth(b_e^t)-1}} + 2\sqrt{|b_e^t|^{-1} \ln t^\alpha} \right)$$

*Proof.* We first prove a lower bound on $\mathbb{E}[c_t^*]$, the expected cost incurred by the optimal algorithm. By definition, $c_t^* = \min_{p \in P_t} \sum_{e \in p} f_e(x_e^t)$, so Lemma A.4, we have $f_e(x_e^t) \geq u_e^t$ with probability at least $1 - t^{-\alpha/\beta^2}$. For brevity, let $\delta = t^{-\alpha/\beta^2}$. Since $c_t^* \geq 0$ always, we have

$$\mathbb{E}[c_t^*] \geq (1 - \delta) \cdot \min_{p \in P_t} \sum_{e \in p} u_e^t$$

Next, we prove an upper bound on $\mathbb{E}[c_t]$, the expected cost incurred by our algorithm. Note that

$$\mathbb{E}[c_t] = \sum_{e \in p_t} f_e(x_e^t) = \sum_{e \in p_t: t \in T_e} f_e(x_e^t) + \sum_{e \in p_t: t \notin T_e} f_e(x_e^t)$$

We will first handle the first term, and then the second term. Since $f_e$ is $L$-Lipschitz and $f_e(0) = 0$, we have $f_e(x) \leq x_{max}^t L$ for all $x \leq x_{max}^t$. Since $x_e^t \leq x_{max}^t$, we have $f_e(x_e^t) \leq x_{max}^t L$ as a broad upper bound for all cases. If $t \in T_e$, then Lemma A.5 implies that $f_e(x_e^t) \leq u_e^t + 2^{1-depth(b_e^t)} x_{max}^t L + 2\sqrt{|b_e^t|^{-1} \ln t^\alpha}$ with probability at least $1 - \delta$. Also recall that $p_t = \arg\min_{p \in P_t} \sum_{e \in p} u_e^t$ by definition. Thus

$$\sum_{e \in p_t: t \in T_e} f_e(x_e^t) \leq \sum_{e \in p_t} \left[ \delta x_{max}^t L + (1 - \delta) \left( u_e^t + \frac{x_{max}^t L}{2^{depth(b_e^t)-1}} + 2\sqrt{|b_e^t|^{-1} \ln t^\alpha} \right) \right]$$

$$\leq \delta |E| x_{max}^t L + (1 - \delta) \cdot \min_{p \in P_t} \sum_{e \in p} u_e^t + (1 - \delta) \sum_{e \in p_t} \left( \frac{x_{max}^t L}{2^{depth(b_e^t)-1}} + 2\sqrt{|b_e^t|^{-1} \ln t^\alpha} \right)$$

$$\leq \delta |E| x_{max}^t L + (1 - \delta) \cdot \min_{p \in P_t} \sum_{e \in p} u_e^t + \sum_{e \in p_t} \left( \frac{x_{max}^t L}{2^{depth(b_e^t)-1}} + 2\sqrt{|b_e^t|^{-1} \ln t^\alpha} \right)$$

Next, for $\sum_{e \in p_t : t \notin T_e} f_e(x_e^t)$ we have

$$\sum_{e \in p_t : t \notin T_e} f_e(x_e^t) \leq \sum_{e \in p_t : t \notin T_e} x_{e,max}^t L = |\{e \in p_t : t \notin T_e\}| x_{e,max}^t L$$

Combining this with our upper bound on $\mathbb{E}[c_t^*]$, we get

$$\mathbb{E}[c_t - c_t^*] \leq |\{e \in p_t : t \notin T_e\}| x_{e,max}^t L + \frac{|E| x_{max}^t L}{t^{\alpha/\beta^2}} + \sum_{e \in p_t} \left( \frac{x_{max}^t L}{2^{\texttt{depth}(b_e^t)-1}} + 2\sqrt{|b_e^t|^{-1} \ln t^\alpha} \right)$$

Summing the second term gives us[4]

$$\sum_{r=1}^t \frac{|E| x_{max}^r L}{r^{\alpha/\beta^2}} = |E| x_{max}^t L \sum_{r=1}^t \frac{1}{r^2} \leq |E| x_{max}^t L \sum_{r=1}^\infty \frac{1}{r^2} = \frac{|E| x_{max}^t L \pi^2}{6}$$

For the first term, we further divide time steps $t \notin T_e$ into (1) time steps where we create a new depth 0 bucket, and (2) time steps where we use edge $e$ for the first time. For case (1), let $Q_e$ denote the set of these time steps. Recall that we only create a depth 0 bucket when the current flow $x_e^t$ does not fall into any bucket we already have. In particular, that implies it is the maximum flow we have seen on this edge: formally, $t \in Q_e$ implies $x_e^t = x_{e,max}^t$. For case (2), let $z_e(t)$ be the indicator variable which takes on value 1 if $t$ is the first time that $e$ is used, and 0 otherwise. Since this happens at most once per edge, we have $\sum_{r=1}^t z_e(r) \leq 1$ for all $e \in E$. Therefore,

$$L \sum_{r=1}^t |\{e \in p_r : r \notin T_e\}| x_{e,max}^r = L \sum_{r=1}^t x_{e,max}^r \left( |\{e : r \in Q_e\}| + |\{e \in p_r : z_e(r) = 1\}| \right)$$

$$\leq L \sum_{r=1}^t |\{e : r \in Q_e\}| x_e^r + L \sum_{r=1}^t x_{e,max}^r \sum_{e \in p_r} z_e(r)$$

$$\leq L \sum_{e \in E} \sum_{r \leq t : r \in Q_e} x_e^r + L \sum_{e \in E} \sum_{r=1}^t x_{max}^t z_e(r)$$

$$\leq L \sum_{e \in E} \sum_{r \leq t : r \in Q_e} x_e^r + |E| x_{max}^t L$$

$$\leq L \sum_{e \in E} 2 x_{max}^t + |E| x_{max}^t L \quad \text{(Lemma A.6)}$$

$$= 2|E| x_{max}^t L + |E| x_{max}^t L$$

$$= 3|E| x_{max}^t L$$

Putting this all together, we get

$$R_t = \sum_{r=1}^t \mathbb{E}[c_t - c_t^*] \leq 3|E| x_{max}^t L + \frac{|E| x_{max}^t L \pi^2}{6} + \sum_{r=1}^t \sum_{e \in p_r} \left( \frac{x_{max}^t L}{2^{\texttt{depth}(b_e^t)-1}} + 2\sqrt{|b_e^t|^{-1} \ln t^\alpha} \right)$$

$$= \frac{|E| x_{max}^t L (\pi^2 + 18)}{6} + \sum_{r=1}^t \sum_{e \in p_r} \left( \frac{x_{max}^t L}{2^{\texttt{depth}(b_e^t)-1}} + 2\sqrt{|b_e^t|^{-1} \ln t^\alpha} \right)$$

as required. □

Let $S_e^t$ be the set of buckets associated with edge $e$ that have existed up to and including time $t$, and let $A_t(b) = \{r \leq t : e \in p_r \text{ and } b_e^t = b\}$ be the set of time steps that we used bucket $b$. We know

---
[4]Note that we could have chosen any $\alpha \geq 2\beta^2$ and simply obtained a different constant.

that every time we use edge $e$ (i.e., $e \in p_t$), we use exactly one bucket $b_e^t$. Thus we can rewrite the bound from Lemma A.7 as

$$R_t \leq \frac{|E|x_{max}^t L(\pi^2 + 12)}{6} + \sum_{e \in E} \sum_{b \in S_e^t} \sum_{r \in A_t(b)} \left( \frac{x_{max}^t L}{2^{\text{depth}(b)-1}} + 2\sqrt{|b_e^r|^{-1} \ln t^\alpha} \right)$$

Note that we use $|b_e^r|$ instead of $|b|$ in order to denote the size of the bucket on time step $r$ specifically.

Let $R_t(b) = \sum_{r \in A_t(b)} \left( 2^{1-\text{depth}(b)} x_{max}^t L + 2\sqrt{|b|^{-1} \ln t^\alpha} \right)$ be the total regret incurred by bucket $b$, and let $R_t(e) = \sum_{b \in S_e^t} R_t(b)$ be the total regret incurred by edge $e$. The rest of the proof is devoted to bounding $R_t(e)$; after we have done so, we can simply sum this bound across all edges.

We next give a brief proof of a standard inequality that will be useful to us.

**Lemma A.8.** *For all $n$, $\sum_{i=1}^n \frac{1}{\sqrt{i}} \leq 2\sqrt{n}$.*

*Proof.* The proof is by induction $n$. The base case of $n = 1$ is trivial, so consider an arbitrary $n > 1$ and assume the claim holds for $n - 1$. Then

$$\sum_{i=1}^n \frac{1}{\sqrt{i}} = \frac{1}{\sqrt{n}} + \sum_{i=1}^{n-1} \frac{1}{\sqrt{i}}$$
$$\leq \frac{2}{\sqrt{n} + \sqrt{n-1}} + 2\sqrt{n-1}$$
$$= 2(\sqrt{n} - \sqrt{n-1}) + 2\sqrt{n-1}$$
$$= 2\sqrt{n}$$

as required. $\square$

We will now bound the maximum regret a single bucket of depth $m$ can contribute.

**Lemma A.9.** *Let $b \in S_e^t$ be a bucket with depth $m$. Then*

$$R_t(b) \leq 4\sqrt{h(m) \ln t^\alpha} + \frac{x_{max}^t L h(m)}{2^{m-1}}$$

*Proof.* Substituting $m = \text{depth}(b)$, $R_t(b)$ is defined by

$$R_t(b) \leq \frac{x_{max}^t L |A_t(b)|}{2^{m-1}} + 2 \sum_{r \in A_t(b)} \sqrt{|b|^{-1} \ln t^\alpha}$$

Next, observe that $|A_t(b)| \leq h(m)$: $|b| = 1$ initially, and once $|b| > h(m)$, we split it into two new buckets and never use it again. Thus we can rewrite the above bound to sum over the number of elements in $b$:

$$R_t(b) \leq \frac{x_{max}^t L |A_t(b)|}{2^{m-1}} + 2\sqrt{\ln t^\alpha} \sum_{i=1}^{|A_t(b)|} \frac{1}{\sqrt{i}}$$
$$\leq \frac{x_{max}^t L h(m)}{2^{m-1}} + 2\sqrt{\ln t^\alpha} \sum_{i=1}^{h(m)} \frac{1}{\sqrt{i}}$$
$$\leq \frac{x_{max}^t L h(m)}{2^{m-1}} + 4\sqrt{h(m) \ln t^\alpha} \quad \text{(Lemma A.8)}$$

as required. $\square$

Next, we will analyze a linear program which will be useful in bounding the total regret contributed by a single edge.

**Lemma A.10.** *Consider the following linear program, parameterized by $t, n$ and $k$:*

$$\max_{x_1,\ldots,x_n \in \mathbb{R}_{\geq 0}} \sum_{m=1}^{n} x_m 2^{km} \tag{1}$$

$$s.t. \sum_{m=1}^{n} 2^{2m-3} x_m \leq t$$

$$x_m \leq 2^m \ \forall m \in \{1, \ldots, n\}$$

*Assume $0 \leq k < 2$, and let $(x_1, \ldots, x_n)$ be an optimal solution, and suppose $x_m > 0$. Then for all $i < m$, $x_i = 2^i$.*

*Proof.* Suppose not, and let $\delta = \min(x_m, 2^i - x_i) > 0$. Define a new solution $(y_1, \ldots, y_n)$ by $y_i = x_i + \frac{\delta}{2^{2i-3}}$, $y_m = x_m - \frac{\delta}{2^{2m-3}}$, and $y_j = x_j$ for $j \notin \{i, m\}$.

We first claim that $y$ is feasible. We have $y_m \geq x_m - \delta \geq 0$, and $y_i \leq x_i + \delta \leq 2^i$. Since $y_j = x_j$ for $j \notin \{i, m\}$, we have $0 \leq x_j \leq 2^j$ for all $j$. Also, we have

$$\sum_{j=1}^{n} y_j 2^{2j-3} = \left(x_i + \frac{\delta}{2^{2i-3}}\right) \cdot 2^{2i-3} + \left(x_m - \frac{\delta}{2^{2m-3}}\right) 2^{2m-3} + \sum_{j \notin \{i,m\}} x_j 2^{2j-3}$$

$$= x_i 2^{2i-3} + \delta + x_m 2^{2m-3} - \delta + \sum_{j \notin \{i,m\}} x_j 2^{2j-3}$$

$$= \sum_{j=1}^{n} x_j 2^{2j-3}$$

which must be at most $t$, since $(x_1, \ldots, x_n)$ is assumed to be feasible.

Finally, we claim that $(y_1, \ldots, y_n)$ has a better objective value. We have:

$$\sum_{j=1}^{n} 2^{kj} y_j - \sum_{j=1}^{n} 2^{kj} x_j = y_i 2^{ki} + y_m 2^{km} - x_i 2^{ki} - x_m 2^{km}$$

$$= 2^{ki} x_i + \frac{\delta 2^{ki}}{2^{2i}} + x_m 2^{km} - \frac{\delta 2^{km}}{2^{2m}} - 2^{ki} x_i - 2^{km} x_m$$

$$= \frac{\delta}{2^{(2-k)i}} - \frac{\delta}{2^{(2-k)m}}$$

Since $2 - k > 0$ and $m > i$, we have $2^{(2-k)i} < 2^{(2-k)m}$. Thus the above expression is strictly positive. Therefore $(y_1, \ldots, y_n)$ has a higher objective value, which contradicts the optimality of $(x_1, \ldots, x_n)$. □

For now, we are primarily interested in $k = 1$. Let $g(t, n)$ denote the maximum value of Program 1 for parameters $t$ and $n$, with $k = 1$. Lemma A.10 implies that for a fixed $t$, $n$ actually does not matter, assuming it is sufficiently large: the $\sum_{m=1}^{n} 2^{2m-3} x_m \leq t$ constraint will be saturated by smaller values of $m$, so any additional variables will always have value 0. Formally, for all $t$, there exists $n_t$ such that $g(t, n_t) = g(t, n)$ for all $n \geq n_t$. Thus we can simply take $n$ to be large enough and write $g(t) = g(t, n_t)$. For example, $n_t = t$ suffices.

The next lemma bounds the total regret contributed each "tree", i.e., a set of buckets that share a depth 0 ancestor. Also, at this point we will substitute in $h(m) = 2^{2m}$.

**Lemma A.11.** *Fix an edge $e$ and time $t$. Let $S \subseteq S_e^t$ be a set of buckets which all have the same depth 0 ancestor. Then*

$$\sum_{b \in S} R_t(b) \leq (4\sqrt{\ln t^\alpha} + 2x_{max}^t L) g(t)$$

*Proof.* Let $x_m$ denote the number of buckets $b \in S$ of depth $m$. Letting $h(m) = 2^{2m}$, Lemma A.9 implies that

$$R_t(b) \leq 4\sqrt{h(m) \ln t^\alpha} + \frac{x_{max}^t L h(m)}{2^{m-1}} = (4\sqrt{\ln t^\alpha} + 2x_{max}^t L) 2^m$$

Thus

$$\sum_{b \in B_e^j} R_t(b) \leq (4\sqrt{\ln t^\alpha} + 2x_{max}^t L) \sum_{m=0}^{n_t} x_m 2^m$$

We next claim that $(x_1, \ldots, x_m)$ is feasible for Program 1. We know that $x_m \leq 2^m$, since within a single binary tree, there are at most $2^m$ nodes of depth $m$. It remains to show that $t \geq \sum_{m=1}^{n_t} 2^{2m-3} x_m$.

Note that all buckets of depth at least 1 come in pairs: spending $h(m-1)$ steps on a bucket of depth $m-1$ creates two buckets of depth $m$. Thus the number of time steps we spent on buckets of depth $m-1$ is exactly $\frac{x_m}{2} \cdot h(m-1) = 2^{2m-2} \frac{x_m}{2} = 2^{2m-3} x_m$. Thus the total number of time steps is bounded by

$$t \geq \sum_{m=1}^{n_t} 2^{2m-3} x_m$$

as required.

Therefore $(x_1, \ldots, x_m)$ is a feasible solution to Program 1, and $\sum_{b \in S} R_t(b)$ is at most $(4\sqrt{\ln t^\alpha} + 2x_{max}^t L)$ times the objective value of Program 1 for $(x_1, \ldots, x_m)$. Since the objective value for $(x_1, \ldots, x_m)$ is at most the optimal objective value $g(t)$, we have $\sum_{b \in S} R_t(b) \leq (4\sqrt{\ln t^\alpha} + 2x_{max}^t L)g(t)$. $\qquad\square$

We are now ready to bound the total regret contributed by a single edge.

**Lemma A.12.** *For all $e, t$, we have*

$$R_t(e) \leq (1 + \log x_{max}^t)(4\sqrt{\ln t^\alpha} + 2x_{max}^t L)g(t)$$

*Proof.* Let $S_1, \ldots, S_q$ be a partition of $S_e^t$ such that each $S_i$ contains exactly one depth 0 bucket, and all buckets in $S_i$ have the same depth 0 ancestor. (i.e., each $S_i$ constitutes a single binary tree). By Lemma A.6, $q \leq 1 + \log x_{max}^t$. Therefore

$$\begin{aligned}
R_t(e) &= \sum_{i=1}^{q} \sum_{b \in S_i} R_t(b) \\
&\leq \sum_{i=1}^{q} (4\sqrt{\ln t^\alpha} + 2x_{max}^t L)g(t) \quad \text{(Lemma A.11)} \\
&= q(4\sqrt{\ln t^\alpha} + 2x_{max}^t L)g(t) \\
&\leq (1 + \log x_{max}^t)(4\sqrt{\ln t^\alpha} + 2x_{max}^t L)g(t)
\end{aligned}$$

as required. $\qquad\square$

We are almost done. One of our last remaining tasks is to bound $g(t)$: the optimal objective value of Program 1.

**Lemma A.13.** *For all $t$, we have $g(t) \leq \frac{2^8}{3} t^{2/3}$.*

*Proof.* Fix a time $t$, and let $(x_1, \ldots, x_n)$ be an optimal solution to Program 1 for parameter $t$. Let $m$ be the minimum integer so that $\sum_{i=1}^{m} 2^{2i-3} 2^i > t$. Then there exists $i \leq m$ so that $x_i < 2^i$: otherwise, $\sum_{i=1}^{m} 2^{2i-3} x_i = \sum_{i=1}^{m} 2^{2i-3} 2^i > t$ by assumption, which would imply that $(x_1, \ldots, x_n)$ is infeasible.

Since there exists $i \leq m$ so that $x_i < 2^i$, Lemma A.10 implies that $x_i = 0$ for all $i > m$. Therefore $g(t) = \sum_{i=1}^{n} 2^i x_i = \sum_{i=1}^{m} 2^i x_i = \sum_{i=1}^{m} 2^{2i}$.

By definition of $m$, we have $t \geq \sum_{i=1}^{m-1} 2^{2i-3} 2^i = \sum_{i=1}^{m-1} 2^{3i-3} \geq 2^{3m-9}$. Therefore

$$3m - 9 \leq \log t$$

$$m \leq \frac{1}{3}\log t + 3$$
$$m \leq \log t^{1/3} + 3$$

Therefore

$$\begin{aligned}
g(t) &\leq \sum_{i=1}^{m} 2^{2i} \\
&\leq \frac{4}{3} 2^{2m} \\
&\leq \frac{4}{3} 2^{2\log t^{1/3}+6} \\
&= \frac{4}{3} \cdot 2^6 \cdot t^{2/3} \\
&= \frac{2^8}{3} t^{2/3}
\end{aligned}$$

as required. $\qquad\square$

Our last lemma analyzes space and time complexity of our algorithm.

**Lemma A.14.** *The space complexity and time complexity of Algorithm 1 on time step $t$ are $O(|E|t^{1/3}\log x_{max}^t)$ and $O\Big(|E|(\log t + \log\log x_{max}^t) + \mathtt{ShortestPath}(|E|,|V|)\Big)$, respectively.*

*Proof.* For the space complexity, note that each bucket $b$ requires a constant amount of space: it simply needs to store $|b|$, $\mathrm{dom}(b)$, $\mathrm{depth}(b)$, and $\mathrm{estimate}(b)$, each of which take constant space. Thus the space complexity is the total number of buckets at time $t$. In fact, we will bound $|S_e^t|$, the number of buckets that have existed at any time up to time $t$ for edge $e$. As in Lemma A.12, let $S_1, \ldots, S_q$ be a partition of $S_e^t$ such that each $S_i$ contains exactly one depth 0 bucket, and all buckets in $S_i$ have the same depth 0 ancestor. We claim that $|S_i| \leq t^{1/3}$ for all $i$.

We will show this using Program 1, but now with $k = 0$ instead of $k = 1$:

$$\max_{x_1,\ldots,x_n \in \mathbb{R}_{\geq 0}} \sum_{m=1}^{n} x_m \tag{2}$$
$$s.t. \sum_{m=1}^{n} 2^{2m-3} x_m \leq t$$
$$x_m \leq 2^m \; \forall m \in \{1,\ldots,n\}$$

The analysis proceeds similarly to Lemmas A.11 and A.13. Let $g'(t)$ be the optimal objective value for Program 2 with parameter $t$. We claim that $|S_e^t| \leq g'(t)$. Let $x_m$ denote the number of buckets of depth $m$ that have existed up to and including time $t$ for this edge. Since the constraints in Program 2 are the same as in Program 1, we already showed in the proof of Lemma A.12 that $(x_1, \ldots, x_n)$ is feasible for Program 2. Since $|S_e^t| = \sum_{m=1}^{n} x_m$, we have $|S_e^t| \leq g'(t)$.

It remains to bound $g'(t)$. Let $m$ be the minimum integer such that $\sum_{i=1}^{m} 2^{2i-3} 2^i > t$. Then as argued in Lemma A.13, $m \leq \log t^{1/3} + 3$. Therefore $g'(t) \leq \sum_{i=1}^{m} x_m \leq \sum_{i=1}^{m} 2^m \leq 2^{m+1}$, so

$$g'(t) \leq 2^{\log t^{1/3}+4} = O(t^{1/3})$$

By Lemma A.6, $q \leq 1 + \log x_{max}^t$. Therefore the total number of buckets for edge $e$ at time $t$ is at most

$$\begin{aligned}
|S_e^t| &= \sum_{i=1}^{q} |S_i| \\
&\leq q \cdot O(t^{1/3}) \\
&= O(\log x_{max}^t t^{1/3})
\end{aligned}$$

and thus the total space complexity is $O(|E| \log x_{max}^t t^{1/3})$.

For the time complexity, each time step involves two tasks that may take non-constant time: computing $u_e^t$ for each $e \in E$, and computing $\arg \min_{p \in P_t} \sum_{e \in p} u_e^t$. Once the first task has been done, the second task can by running any shortest path algorithm on the directed graph $(E, V)$ where edge $e$ has nonnegative weight $u_e^t$; this takes time $\texttt{ShortestPath}(|E|, |V|)$.

For the first task, there are two parts that may take non-constant time. The first part is searching for a bucket $b$ such that $y \in b$. The buckets correspond to non-overlapping intervals, so this can be done by keeping the buckets in sorted order and using binary search. This approach yields a time complexity of $O(\log |S_e^t|) = O(\log t + \log \log x_{max}^t)$ per edge and thus $O(|E|(\log t + \log \log x_{max}^t))$ total. Note that to keep the buckets in sorted order, creating a bucket now requires us to insert it into the right place in the ordering. Using a self-balancing binary tree, this be done in time $O(\log |S_e^t|)$ per operation as well (and at most two buckets are created per edge per time step).

The second part that may take non-constant time is computing $y_{max} = \max \left( \cup_{b \in B_e} \text{dom}(b) \right)$, if applicable. If the buckets are stored in sorted order, this can be done in constant time by looking at the last bucket in the ordering. This yields the desired bound. □

We are finally ready to prove Theorem 1.1.

**Theorem 1.1.** *The expected regret of Algorithm 1 after $t$ time steps is*

$$R_t = O\left( t^{2/3} \cdot |E| \log x_{max}^t (\beta \sqrt{\log t} + x_{max}^t L) \right)$$

*The space complexity and time complexity on time step $t$ are $O(|E| t^{1/3} \log x_{max}^t)$ and $O\left( |E|(\log t + \log \log x_{max}^t) + SP(|E|, |V|) \right)$, respectively.*

*Proof.* We have

$$R_t \leq \frac{|E| x_{max}^t L(\pi^2 + 12)}{6} + \sum_{e \in E} R_t(e) \quad \text{(Lemma A.7)}$$

$$\leq \frac{|E| x_{max}^t L(\pi^2 + 12)}{6} + \sum_{e \in E} (1 + \log x_{max}^t)(4\sqrt{\ln t^\alpha} + 2x_{max}^t L)g(t) \quad \text{(Lemma A.12)}$$

$$= \frac{|E| x_{max}^t L(\pi^2 + 12)}{6} + |E|(1 + \log x_{max}^t)(4\sqrt{\ln t^\alpha} + 2x_{max}^t L) \cdot \frac{2^8}{3} t^{2/3} \quad \text{(Lemma A.13)}$$

Plugging in $\alpha = 2\beta^2$ as in Lemma A.7 gives us

$$R_t \leq \frac{|E| x_{max}^t L(\pi^2 + 12)}{6} + |E|(1 + \log x_{max}^t)(4\beta\sqrt{2 \ln t} + 2x_{max}^t L) \cdot \frac{2^8}{3} t^{2/3}$$

$$= O\left( |E| \log x_{max}^t (\beta \sqrt{\log t} + x_{max}^t L) t^{2/3} \right)$$

Combining this with Lemma A.14 proves the theorem. □

