# OpenReview forum: "Robust Learning for Congestion-Aware Routing"
_ICLR.cc/2021/Conference — Reject_

### Official Review · AnonReviewer3 · 2020-10-24
**Interesting (theoretical) algorithm**

**Rating:** 8
**Confidence:** 4

**Review:**

In this submission a routing problem is studied. In the considered model with each edge of the given graph a congestion function is associated that specifies the congestion depending on the current load of the edge. Then cars have to be routed through the network where each car has a source and a destination and one aims at choosing a path from the source to the destination with the smallest total congestion. However, the congestion functions of the edges are a priori unknown and hence one cannot trivially use a shortest path algorithm. Instead one gains information about the congestion functions only by routing the cars. When a car is routed one observes for each edge on its path the current congestion up to some random additive term. These observations can then be used for future routing decisions.

The main result of the submission is an algorithm that achieves a cumulative sublinear regret of O(|E|t^{2/3}) where the regret is defined as the difference between the expected path length chosen by the algorithm and the length of the shortest path. Some experiments are also conducted with this algorithm but the focus of the submission is clearly on the theoretical results.

The algorithm itself is non-trivial but also not too surprising. Due to the random noise one needs enough samples to estimate the congestion on an edge. The difficulty is that the congestion depends on the current load, i.e., one needs enough samples close to the current load (here one uses that the congestion functions are assumed to be Lipschitz continuous). The algorithm cleverly and adaptively partitions the samples of an edge into buckets where each bucket represents a certain range of loads. Then there are two factors that determine the precision of the congestion estimation: the range of the buckets and the number of samples per bucket. The more samples one has in a certain range the smaller can the ranges of the buckets be. Basically the algorithm splits a bucket if it contains already enough samples.

I find the results interesting. The proposed algorithm is natural and its analysis is non-trivial. I am not completely sure if the model is very realistic though. It is assumed that the congestion functions are unknown (which makes sense). However, if I understand it correctly it is assumed that each driver knows exactly the current load on all the edges. It is not clear to me why this makes sense. It might also be the other way round: While the congestion functions are more or less known from historic data, the current load is unknown to the drivers. In my opinion the authors could discuss the model in more detail.

---

> ### Author Response · Authors · 2020-11-17
> **Response to Reviewer 3**
>
> We thank the reviewer for the constructive response. The reviewer’s main concern is the following:
>
> > I am not completely sure if the model is very realistic though. It is assumed that the congestion functions are unknown (which makes sense). However, if I understand it correctly it is assumed that each driver knows exactly the current load on all the edges. It is not clear to me why this makes sense. It might also be the other way round: While the congestion functions are more or less known from historic data, the current load is unknown to the drivers. In my opinion the authors could discuss the model in more detail.
>
> From the viewpoint of a navigation app, the location of each car using the app is known, and thus the number of cars on an edge that are using the app is known. Of course, not every car is using the app, but if the app has a good estimate of the percentage of cars that tend to use it, a good estimate of the current load should still be possible.
>
> While our model does assume that the algorithm knows exactly the current load on each edge, the bucketing approach of our algorithm should be robust to small inaccuracies in the load. Even if the inaccuracy leads to the algorithm using a different bucket, this error decreases as the length of each bucket shrinks.
>
> Inspired by this reviewer's concern, we have run additional experiments where the algorithm only sees a noisy version of the load (with the optimal algorithm still seeing the exact load). The results indicate that the algorithm is indeed robust to inaccuracy in the load, at least empirically. Specifically, we see a sublinear regret curve similar to the results of the original experiments (where the algorithm always saw the exact load). Supporting our observations with theoretical analysis is an interesting direction for future work.

---

### Official Review · AnonReviewer4 · 2020-10-27
**An interesting extension of combinatorial semi-bandits**

**Rating:** 7
**Confidence:** 3

**Review:**

This work introduces an interesting generalization of stochastic combinatorial semi-bandits for routing in a static graph. The main differences are: (1) the expected loss of an edge e is f_e(x^t_e) where the flow x^t_e is revealed at the beginning of each round (for each edge) and f_e is an unknown Lipschitz function (with known Lipschitz constant); (2) the regret is dynamic, computed against the sequence of optimal paths. When f_e is a constant function for each edge, then we recover a version of the stochastic combinatorial semi-bandit.

The main contribution is a novel UCB-like algorithm with a dynamic regret bound after T steps of |E|T^{2/3} (ignoring log factors in T). This is larger than the rate |E|T^{1/2} achievable for *adversarial* combinatorial semi-bandits, but ---as we said--- the problem studied here is more general.

The algorithm uses a hierarchical and dynamical bin structure to produce convergent estimates of f_e(x) for different values of x. This is a nice idea which is not standard in the bandit literature. The analysis of the algorithm is quite involved and apparently novel for the most part.

The main ideas behind the analysis are well explained at an intuitive level.

The source of the T^{2/3} dependence should be independent of the combinatorial nature of the semi-bandit problem. It would be interesting to know what happens in the simpler setting of parallel edges. Can the upper bound be improved? And if not, can a tight lower bound be proven?

The definition of x_{max}^t on page 2 looks wrong because the max is over t.

---

> ### Author Response · Authors · 2020-11-17
> **Response to Reviewer 4**
>
> We thank the reviewer for the constructive comments and interesting ideas for extending our work which we will take into account while preparing the next version of our paper.

---

### Official Review · AnonReviewer2 · 2020-10-28
**The paper studies online learning for routing in a city network (a graph) from adversarially given source to destinations and under adversarially given current flow on each road segment (edges), for a fixed but unknown congestion function.**

**Rating:** 3
**Confidence:** 4

**Review:**

The paper uses the bandit learning framework to study the online learning problem for routing in a city network . After each routing decision, the learning agent observes the actual delay on each edge, which is given by the congestion function on the given flow plus a random noise, and the reward is the total delay on all edges. The paper proposes a learning algorithm similar to the UCB approach, provide the regret bound result, and conduct simulations on the New York City network to verify performance of the algorithm.

The contribution of the paper in my view is mainly on the setting where the nonlinear congestion function (satisfying a Lipschitz condition) with arbitrarily given flow is considered, and on using dynamic splitting to refine the budgets based on the number of observations.

However, there are a number of issues in the paper that makes the overall contribution questionable, as I list below.

- The first issue is that the authors is missing an entire line of very relevant result work and results. In particular, the work is closely related to the combinatorial semi-bandit research. The following are several most relevant studies, while many others are available in the literature.

[1] Chen, Wang, Yuan, and Wang. Combinatorial Multi-Armed Bandit and Its Extension to Probabilistically Triggered Arms. JMLR'2016, Conference version appeared in ICML'2013

[2] Qin, Chen, and Zhu. Contextual Combinatorial Bandit and its Application on Diversified Online Recommendation. SDM'2014

The routing problem in the current paper has a number of similarities with the CMAB framework in [1]: each road segment (edge) is a base arm, and it can be individually observed when the selected path contains the edge --- semi-bandit feedback, and each action is a set of base arms (a path) that follows certain constraints. The adversarial chosen path source and destination and the current flow can be viewed as the context in CMAB. The contextual CMAB problem is studied in [2]. The current paper has a bit more complication due to the congestion function. But the authors treat it by discretizing the flow into buckets. Essentially this is treating the pair of (edge, flow budget) as a base arm in the CMAB framework. Therefore, I believe that if we do not do dynamic bucketing and use a fixed discretization budget, the current problem fits as a special instance in the CMAB framework, and thus can be solved by the CUCB algorithm for CMAB. Noticeably the CUCB algorithm has O(\sqrt{t}) regret (ignoring the additional log t term). However, the current paper only has a result for O(t^{2/3}), and the authors mention both in the introduction and conclusion that achieving O(\sqrt{t}) regret as a future research work. But with proper setup as the above, I believe the O(\sqrt{t}) regret has been achieved. Therefore the authors are seriously missing some existing work in this regard. Of course, the above discussion only uses static buckets. But applying dynamic buckets should only improve the results. The static bucketing may give some extra factor in the regret bound in terms of the number of budgets, but that is independent of time t. Therefore, I believe the dependency on time t should be O(\sqrt{t}), and it should be easily achieved under the CMAB framework, such as using the CUCB algorithm in [1].

- The second issue is that in terms of the theoretical analysis, the authors only provide the regret bound result as a theorem in the introduction. There is no more discussion on the factors related to the regret bound, such as |E|, and \beta. Are their dependency tight or not? There is also no explanation on the outline of the analysis. In particular, how to incorporate the dynamic splitting of the buckets into the analysis. Dynamic splitting is the only thing that is different from existing work in my view, and its advantage should be further discussed.

- The third issue is on the experimental evaluation. The authors do not compare the proposed algorithm with any baseline algorithms. Baseline algorithms that could be considered include epsilon-greedy algorithms and Thompson Sampling based algorithms. Without such comparison, it is hard to understand the benefit of the proposed algorithm.

- Another issue is that the proposed algorithm is essentially very close to using the lower confidence bound (LCB) for the edge delays in the CMAB setting. Since the optimization problem is minimizing the delay, so it is understandable that the standard UCB is replaced by the LCB. The authors need to discuss whether there is any more difference between their algorithm and the LCB/UCB based algorithms.

---

> ### Author Response · Authors · 2020-11-17
> **Response to Reviewer 2 (2/2)**
>
> ## Responses to the reviewer's other comments
>
> “The second issue is that in terms of the theoretical analysis, the authors only provide the regret bound result as a theorem in the introduction. There is no more discussion on the factors related to the regret bound, such as $|E|$, and $\beta$. Is their dependency tight or not?”
>
> Please see the response to Reviewer 1 on this topic.
>
> > There is also no explanation on the outline of the analysis. In particular, how to incorporate the dynamic splitting of the buckets into the analysis. Dynamic splitting is the only thing that is different from existing work in my view, and its advantage should be further discussed.”
>
> Space constraints limited our ability to provide an outline of the analysis in the main body. Sections 2.1 and 2.2 do provide some intuition about this analysis, but we plan to further emphasize dynamic splitting in the next revision. For example, we will discuss how dynamic splitting gives us an increasingly fine-grained discretization of the flow, leading to perfect cost estimation in the limit. Dynamic splitting also allows us to carefully balance the number of samples in a bucket with the length of the bucket (i.e., the range of flows it covers), which is crucial for achieving sublinear regret.
>
> > The third issue is on the experimental evaluation. The authors do not compare the proposed algorithm with any baseline algorithms. Baseline algorithms that could be considered include epsilon-greedy algorithms and Thompson Sampling based algorithms. Without such comparison, it is hard to understand the benefit of the proposed algorithm.
>
> Since our work is not a special instance of CMAB, we are not aware of any other algorithms that yield sublinear regret for our problem. Standard CMAB algorithms like epsilon-greedy and Thompson sampling would lead to linear regret in our model, since they essentially treat the congestion functions as constant. Consequently, it is not clear if any prior algorithms would be appropriate for an experimental baseline.
>
> > Another issue is that the proposed algorithm is essentially very close to using the lower confidence bound (LCB) for the edge delays in the CMAB setting. Since the optimization problem is minimizing the delay, so it is understandable that the standard UCB is replaced by the LCB. The authors need to discuss whether there is any more difference between their algorithm and the LCB/UCB based algorithms.
>
> As the reviewer noted earlier, the crucial distinction here is the bucketing approach, and in particular, the dynamic splitting. This allows us to handle the parameterization of the congestion function, as discussed above. We will make this comparison clear in the paper.

---

> ### Author Response · Authors · 2020-11-17
> **Response to Reviewer 2 (1/2)**
>
> We thank the reviewer for the thoughtful response. We first explain why our model is not a special case of combinatorial multi-armed bandits (CMAB), and then address the rest of the reviewer’s comments.
>
> ## Comparison to Combinatorial Multi-Armed Bandits
>
> The reviewer suggests that our work is largely subsumed by the CMAB model. Though we thank the reviewer for pointing us to this line of work, which we will discuss in the related work section of our paper, we explain why this is not the case:
>
> Specifically, the reviewer suggests that our work can be reduced to CMAB if we use static bucketing, but dynamic bucketing is crucial: in particular, the number of buckets increases over time so that we have an increasingly fine-grained discretization of the flow. As the number of buckets approaches infinity, the cost estimation approaches perfect accuracy. As a result, the average regret per time step approaches 0, yielding sublinear cumulative regret.
>
> In contrast, static bucketing can never achieve sublinear regret in our model. Intuitively, this is because cost estimation can have error proportional to the length of the bucket (length refers to the size of the interval corresponding to the bucket). Thus we need the length of each bucket to go to 0 in order to achieve zero average regret per time step.
>
> For a concrete example, consider a graph of two parallel edges with $\beta = 0$ (i.e., zero noise). Suppose edge 1 has congestion function $f_1(x) = xL$ and edge 2 has congestion function $f_2(x) = c$ for some constant c. Let $[y,z]$ be the interval of a static bucket for edge 1, and suppose all of the samples in the bucket correspond to flow $z$. When asked to estimate $f_1(y)$, the algorithm will always return $zL$, when the true cost is $yL$. Suppose $c = zL - \varepsilon$: the algorithm will always choose edge 2, when in reality edge 1 has a cost of $yL < zL - \varepsilon$. In particular, this incurs a regret of $(z-y)L + \varepsilon$ per time step, leading to linear cumulative regret.
>
> Given that dynamic bucketing is necessary, the reviewer’s proposed reduction to CMAB will not work, since the standard CMAB model requires the set of arms to be fixed.
>
> To conclude, one can view our problem as a generalization of CMAB where each arm’s reward distribution takes a real number parameter as input (in our case, the flow on the edge). As reviewer 4 correctly notes, “When $f_e$ is a constant function for each edge, then we recover a version of the stochastic combinatorial semi-bandit.”

---

> > ### Comment · AnonReviewer2 · 2020-11-17
> > **static bucketing + existing result on CUCB is enough to achieve the same (or even better) sublinear regret**
> >
> > The authors provide some explanations, from which I can understand now that using static bucketing may not be able to achieve $\sqrt{T}$ regret. But the authors explanation and the example showing that static bucketing leads to linear regret is incomplete. In fact, by properly choosing the size of the budget (which is dependent on the time horizon $T$), we can achieve sublinear regret. I will show below that using the existing result on combinatorial semi-bandit and its CUCB algorithm with static bucketing, we can achieve $\tilde{O}(T^{2/3})$ regret, and the dependency on other parameters seem to be also better than what the authors obtain in their Theorem 1.1.
> >
> > I will also list a few other issues in the paper and in authors' rebuttal after my explanation on using CUCB below.
> >
> > 1. On achieving $\tilde{O}(T^{2/3})$ regret using CUCB and static bucketing
> >
> > Suppose, on every edge, we set bucket size to be $\varepsilon$, so the total number of buckets is $|E| \cdot x^T_{max} / \varepsilon$. We can treat each bucket as a base arm in the CMAB framework. The action in each round is a path selected from the source and destination, but since in each round each edge has a flow, the actual action is the set of buckets on the path that correspond to the flows on the edges of the path. The flow and the source and destination are given by the adversarial environment, not chosen by the learning agent. This is only difference from the standard CMAB model, but it is straightforward to verify that by considering flows together with the path will give the correct action as the subset of budgets on the path, and the regret analysis is not affected. The reward function is a linear summation, so it belongs to the linear function case. This case is solved with tight upper bound first in the following paper:
> >
> > Kveton, Wen, Ashkan, and Szepesvári. Tight regret bounds for stochastic combinatorial semi-bandits. AISTATS, 2015
> >
> > The following paper provides a simpler analysis and slightly better regret bound for the constant term in their Theorem 4 in the appendix.
> >
> > Wang, Chen. Improving Regret Bounds for Combinatorial Semi-Bandits with Probabilistically Triggered Arms and Its Applications. NIPS'2017
> >
> > Using Theorem 4 in the above paper, it states that the regret bound of CUCB is
> >
> > $O(B\sqrt{KmT \ln T})$,
> >
> > where for the linear function the bounded smoothness constant $B=1$, $K$ is the maximum number of base arms in an action, which is at most $|E|$ in the current setting, $m$ is the total number of base arms, which is $|E| \cdot x^T_{max} / \varepsilon$. We set $\beta=1$ for simplicity. Because of the discretization error (bucketing error), each bucket may has an estimation error of $L \varepsilon$, and each action in each round may have a total error at most $|E| \cdot L \varepsilon$. Thus, the total regret together is:
> >
> > $O(\sqrt{|E|^2 \cdot x^T_{max} \cdot T \ln T / \varepsilon} + |E| \cdot L \cdot T \varepsilon)$.
> >
> > The authors in the rebuttal essentially are trying to claim that the second term above leads to a linear regret. But in fact, we can set $\varepsilon$ to any value we want. So the best value is to make the above two terms equal, which implies setting $\varepsilon = (x^T_{max} \ln T / (L^2 T))^{1/3}$. This leads to the final regret as:
> >
> > $O(T^{2/3} \cdot |E| \cdot (L x^T_{max} \ln T)^{1/3})$.
> >
> > Comparing to the result in Theorem 1.1 in the current submission, the above result matches $T^{2/3} \cdot |E|$ and is better in the term for $L$ and $x^T_{max}$ and $\ln T$ (the above has the exponent of $1/3$ for these terms while Theorem 1.1 has the exponent of $1$ on these terms).
> >
> > Note that the above setting of $\varepsilon$ requires knowing $T$ in advance. But the standard doubling trick could guess $T$ with doubling size to achieve the same regret bound without knowing the actual time horizon $T$.
> >
> > The above static discretization with proper discretization bucket size setting has been used in other bandit related settings, for example in dynamic pricing, as in the following paper:
> >
> > Kleinberg, Leighton. The Value of Knowing a Demand Curve: Bounds on Regret for On-line Posted-Price Auctions. FOCS, 2003.
> >
> > In summary, applying the standard CMAB framework with a standard (static) discretization technique is sufficient to solve the proposed problem, with a matching (or even better) regret upper bound. Therefore, it is unclear on the contribution of the paper, and the need of the dynamic bucketing proposed in the paper.

---

> > > ### Comment · AnonReviewer2 · 2020-11-17
> > > **Further comments on the paper and the rebuttal**
> > >
> > > 2. Further comments on the paper
> > >
> > > It is unclear what is the exact definition of $x^t_{max}$. As the reviewer 4 pointed out, the definition in Page 2 seems to be incorrect, since it is taking max over $t$ while it is on a fixed $t$. I am guessing that it should be the maximum values of all flows that appear by round $t$. But whatever the definition, there is an issue with its appearance in Theorem 1.1. In this theorem, $\log x^t_{max}$ appears both in the regret bound and in the time and space complexity. But what prevents $x^t_{max}$ to take the value of $1$? If so, $\log x^t_{max} = 0$ and all the results in Theorem 1.1 collapse. Some explanation and clarity are needed here.
> > >
> > > 3. Further comments on the rebuttal
> > >
> > > In the parallel edge example the authors give in the rebuttal,  it first assumes that "all of the samples in the bucket correspond to flow $z$." Then it says that flow $y$ is given and we are asked to estimate $f_1(y)$. If $y$ appears for the first time, it is true that the algorithm will return the estimate as $zL$, and under their condition, the algorithm will select edge 2 but the best choice should be edge 1 with cost $yL$. First, in this case, the regret should be $c - yL$, not $(z-y)L + \varepsilon$ as stated in the rebuttal. Second, this is only true when $y$ appears for the first time as the flow for edge 1. When $y$ appears more time, the estimate on edge 1 will not longer be $zL$ and the feedback for flow of $y$ should be considered. So by itself it is not always the case that we will always select the incorrect second edge and incur a linear regret. Of course, this is not the main point, and as I discussed above, I acknowledge that if we treat bucket size $\varepsilon$ as a constant, then we could have a linear regret of $\varepsilon |E| L T$, but we are allowed to set $\varepsilon$ to be inversely related to $T$ to achieve a sublinear regret.

---

> > > > ### Author Response · Authors · 2020-11-19
> > > > **Second Response to Reviewer 2**
> > > >
> > > > ## Summary of our response
> > > >
> > > > The reviewer proposes an alternative solution to our problem based on prior work in the CMAB model. Although it is possible that the reviewer’s ideas could yield an alternative algorithm for our problem with similar regret bounds to our own, there are crucial differences between our model and those in the cited papers that preclude a straightforward black box application of previous results. We elaborate on this below. The reviewer’s proposal is far from a complete proof, and its correctness hinges on addressing various details regarding the aforementioned differences. For instance, it should be noted that the reviewer has already backtracked their initial assertion that static buckets are sufficient to obtain a $O(\sqrt{T})$ regret, upon delving into the details of the bucketing.
> > > >
> > > > Furthermore, the reviewer’s new proposal incorporates many of the conceptual ideas and techniques that we advance in our work (in particular, shrinking bucket sizes), which form a crucial part of this submission’s contributions. In this sense, the reviewer’s comments may actually reinforce the value of our conceptual contributions.
> > > >
> > > > Overall, independent of the validity of the reviewer’s approach, we believe that our submission should be considered by the ICLR committee based on its own merits.
> > > >
> > > > Before delving into the details of the reviewer’s proposal, we also note that our algorithm has a space complexity advantage over the reviewer’s proposed algorithm. Our dynamic bucketing approach enables us to avoid storing all observations by judiciously combining observations into common buckets, thereby leading to a space complexity of O(T^(1/3)). This is not immediately possible in the reviewer’s approach since a given observation can change buckets as the discretization changes. Thus the reviewer’s proposal seems to incur a linear space complexity. It is possible that this could be remedied, but like many details of the reviewer’s approach, this is not addressed.
> > > >
> > > > ## Elaboration
> > > >
> > > > We now elaborate on why the reviewer’s approach is not a straightforward application of the prior CMAB results.
> > > >
> > > > 1. The fact that the combinatorial actions change over rounds in our setting implies that the relevant work discussed in the reviewer's outline cannot apply as a black box. Specifically, in the reviewer's approach, the relevant bucket for each edge would change between time steps and thus not all observations from the edge can be used in the estimation. The previous work cited does not need to handle such issues. The reviewer claims that “it is straightforward to verify that by considering flows together with the path will give the correct action as the subset of budgets [buckets?] on the path, and the regret analysis is not affected”. We believe this is a crucial difference and should not be glossed over. In fact, this approach may even necessitate reworking many of the lemmas and theorems from the cited papers in order to verify that they still work in our setting.
> > > >
> > > > 2. The reviewer claims that our discretization approach is a standard technique which has been applied in other bandit settings, citing a FOCS 2003 paper on dynamic pricing. However, the FOCS 2003 paper has an action space that is simply a set of discretized prices out of which the auctioneer needs to pick one. This is in stark contrast to our dynamic discretization and we believe it is not a reasonable parallel, as it is a much simpler scenario.
> > > >
> > > > 3. We also observe that the reviewer's proposal uses a doubling trick to guess the horizon T and that this trick leads to buckets whose sizes decrease over time. This is the exact same insight motivating our dynamic bucket splitting approach: we use an increasingly fine-grained discretization to eliminate regret.
> > > >
> > > > To summarize, the reviewer provides a sketch of an alternative approach to solving our problem. Several important issues are glossed over, and we believe this argument requires a formal proof. Such sketches often fail in the details, as was for example the case with the original suggestion (which would indeed be simpler and different from our approach if it had worked). Furthermore, the reviewer’s new approach shares several crucial features with our algorithm, and seems to have been designed using novel insights we provide in our work (e.g., shrinking bucket sizes). In general we feel that discussing other potential designs and trying to validate or refute them is steering the conversation away from our work and is not relevant to the merit of our paper.

---

> > > > > ### Comment · AnonReviewer2 · 2020-11-19
> > > > > **Further clarification (1/3)**
> > > > >
> > > > > 1. I disagree with the authors' statement that "independent of the validity of the reviewer’s approach, we believe that our submission should be considered by the ICLR committee based on its own merits." If a reviewer finds that applying a couple of standard techniques known in the field can easily achieve the same (or even better) result that the authors propose, it is very reasonable that the authors should provide serious comparisons with this approach, and the paper should be evaluated based on such comparisons, since many readers could also independently come up with the same simple approach and questioning the validity and novelty of the new approach in the paper. ignoring such simple and alternative approach and evaluating the paper's contribution independent of this alternative simple approach is a disservice to the review process, and is exactly what we should avoid in the review process.
> > > > >
> > > > > 2. I am not saying that the proposed approach in the paper has no contribution. The dynamic bucketing approach may be quite an interesting and useful contribution. However, it should be evaluated under the right context, which means that it needs to be compared against a simple alternative approach that I propose (and many others in this field may likely propose). If the authors put a serious effort in comparing these two approaches, the paper may have its merit for publication in a good venue. But in the current form, without such a comprehensive comparison, the paper is not ready for acceptance at ICLR.
> > > > >
> > > > > 3. One key point is on whether the method I proposed is simple and standard. So I add a couple of more points here. My method is based on two existing techniques, a) combinatorial semi-bandit (CSB) approach, and b) static discretization related to time horizon T. First, for combinatorial semi-bandit, as I already pointed out in my previous comments, it has been around for quite a while, with rich research results that are very relevant to the current work. But the paper completely ignores this line of research. Second, the static discretization related to time horizon T, this has also been around for quite a while, as a reference I gave in my previous response. On this point, I also need to clarify that the authors may have misunderstood me in their elaboration response item 2. I am not saying that their dynamic discretization is in parallel of the earlier work such as the one in the FOCS'2003 paper. I am saying that a different discretization, namely static discretization, and the size of each bucket is set to be inversely related to the time horizon T, has been around and is a standard way to deal with continuous variables. I am saying that applying this standard discretization (again different from the dynamic discretization proposed in the paper) together with CSB is sufficient to achieve a regret bound that is even better than what is shown in the paper.

---

> > > > > > ### Comment · AnonReviewer2 · 2020-11-19
> > > > > > **Further Clarification (2/3)**
> > > > > >
> > > > > > 4. The authors pointed out that my approach using CSB is not black-box reduction, since it may need some extension to the standard CSB work, and some details are not fully given. Here, I would not argue it is easy or not, correct or not, although I believe it is correct and easy to verify. My point is that, as a reviewer, I am giving reasonable suggestions to the authors on some simple alternative approaches. I believe this is how a constructive reviewing process normally goes --- the reviewer provides constructive comments and ideas to improve the paper, and the authors should further investigate the proposals and make their assessments and conclusions, and include their investigations in the revised paper as an improvement. Thus, it is up to the authors to verify that if my proposed approach is indeed simple and correct, and it should not be me as a reviewer to fill in every detail or give a full proof to the authors. In the first round, I already suggested that the authors should look into the CSB framework and make comparison and see if CSB could achieve the goal in a much simpler way. But instead of investigating the CSB approach in detail, the authors seem to be eager to dismiss this suggestion, and simply say that CSB is not that relevant, and that the static discretization would lead to linear regret. Then, in my reply to this, I have to give much more detail to show that CSB is indeed relevant, and discretization with proper setting of bucket size will have sublinear regret, in fact achieving a regret better than their result in some terms. But the authors still come with push back saying my argument is not a full proof and some details are missing. The authors further try to discredit my comment by pointing out that I backtracked from my initial assessment that one could achieve $O(\sqrt{T})$ approach. I think the authors is not taking my detailed constructive comments in a positive way, and only trying to argue a way out for their paper at this point. My focus on suggesting $O(\sqrt{T})$ initially is trying to emphasize that CSB might be a liable approach that already achieve very good result. At that point I haven't thought more about the impact of discretization so I suggest that it might be giving $O(\sqrt{T})$ regret. Even though it is not so, and in the second round I corrected it but saying that CSB is still very useful to give a $O(T^{2/3})$ regret with other terms matching with or better than the terms in the current paper. Again, I believe it is the authors' responsibility to take the reviewer's suggestions constructively and seriously check the suggestions from the reviewers and evaluate if it is indeed correct, and provide reasonable comparisons in the rebuttal and in the revision of the paper.
> > > > > >
> > > > > > 5. The authors do not respond to my comment in my previous reply, showing that the regret in my approach is better than theirs. In particular, the regret of my approach is $O(T^{2/3} \cdot |E| \cdot (L x^T_{max} \ln T)^{1/3})$, while their result is $O(T^{2/3} \cdot |E| \cdot L x^T_{max} \ln T)$, and thus their result is worse in $L$, $x^T_{max}$ and $\ln T$ terms.
> > > > > >
> > > > > > 6. In terms of of the space complexity, the authors mention in their reply that their algorithm achieves a space complexity of $O(T^{1/3})$, while my approach may have space complexity linear in $T$. Again, this is incorrect. As I have explained in my previous reply, given time $T$, the total number of base arms in my approach is $|E| x^T_{max} / \varepsilon$, and with the $\varepsilon = (x^T_{max} \ln T / (L^2 T))^{1/3}$ as given in my previous reply, we can see that the total number of base arms is $O(T^{1/3})$. Each base arm only needs a constant space to update its frequency and estimate in CSB. So the total space complexity in my approach is $O(T^{1/3})$, also matching their approach. I do not understand why the authors say that "This is not immediately possible in the reviewer’s approach since a given observation can change buckets as the discretization changes". In my approach, observation will not change buckets, and I am using static bucketing (for a given $T$).

---

> > > > > > > ### Comment · AnonReviewer2 · 2020-11-19
> > > > > > > **Further Clarification (3/3)**
> > > > > > >
> > > > > > > 7. Finally, in terms of the doubling trick to handle unknown $T$, this is again standard practice in many MAB studies. The authors correctly point out that doubling $T$ will have the effect of dynamic (or periodically) shrinking the bucket size, and this has similarity with dynamic bucketing proposed in the current paper. However, the assessment that my approach "seems to have been designed using novel insights we provide in our work (e.g., shrinking bucket sizes)", is completely incorrect. I am using standard static bucketing for a given $T$, and then using the standard doubling trick to handle unknown $T$, and it has nothing to do with the insight obtained by the authors' work. On the contrary, the authors should provide another serious comparison between their dynamic bucketing and the existing static bucketing with given $T$ plus the doubling $T$ with unknown $T$, which also leads to dynamic bucketing. The latter is clearly prior art of the authors' work, not the other way round. By the way, the doubling trick also appears in the same FOCS'03 paper, and it has been used in many other studies.

---

> > > > > > > > ### Author Response · Authors · 2020-11-19
> > > > > > > > **Third Response to Reviewer 2**
> > > > > > > >
> > > > > > > > Our thesis is that in this thread we are discussing a parallel design that has been constructed after observing our solution, specifically the way we modify arms and use shrinking buckets. This naturally points to previous work that does parallel things and we are happy the reviewer pointed this work to us for discussion in the paper, as we previously said. The design proposed uses modifications of prior work in a non-obvious way and we were not able to verify it after spending a significant amount of time reading the cited papers. We are certainly not dismissing the reviewer's suggestions, a comment we consider unfair, since we have agreed to discuss the papers in the related work section, we have refuted the first suggestion, and we have thoroughly investigated the second one to the extent that the little time available permits.

---

### Official Review · AnonReviewer1 · 2020-10-29
**Nice contribution**

**Rating:** 5
**Confidence:** 4

**Review:**


In this paper the authors study the problem of routing users according to their requests through a network with unknown congestion functions over infinite time horizon. They model the problem as follows. A directed graph G(V,E) is given, where each edge is associated with  a congestion function f_e that maps the flow along edge e to some positive real. The mild assumption the authors make on the congestion functions of all edges is that they are L-Lipschitz. At every timetick a car enters the routing app, and asks to move from a source to destination. Given the collection of paths, the routing app is required to make a choice of a path. This choice incurs a cost, which is the noisy version of the sum of the congestions along the edges of the path Again, the author(s) make the reasonable assumption that the noise is zero mean and bounded by some value beta. The model is nicely motivated by real-world aspects of routing apps, and is a clean mathematical model. The key result is stated as Theorem 1.1. I checked the proof, and it appears solid. The authors design algorithm 1. ITs intuition is well described in section 2.1 Some comments to the authors of the paper follow


- Is the Lipschitz constant known? If not, is there a way to test this assumption?
- I think the authors mean that the regret R_t is \sum_{r=1}^t E[c_r-c*_r].
- Can you elaborate on your conjecture (e.g., page 3) concerning the right asymptotics of the regret as a function of t?  Is your conjecture related to the Awerbuch-Kleinberg SODA paper?
- Is the dependence of |E| tight? It felt while reading the proof (e.g., in Lemma A.7 where the summation over time and edges along the path are exchanged) that perhaps the analysis could be improved.
- The experimental part is the weakest part in this paper. I would urge the authors to try settings where the assumptions of the theorem start breaking down, and see how the regret changes. For instance, what if some congestion functions are not L-Lipschitz but all the rest are. What if these functions correspond to edges with  high betweenness centrality? One can think of many other settings that could have made the paper (including the supplementary material) more interesting

Overall this is a well-written paper with a clear contribution. The weak parts are the experiments which are practically sanity check for the validity of theorem 1.1, and the fact that from a technical point of view there is not much novelty. The novelty in my opinion lies in the design of the estimation algorithm of the cost.

[Score updated From 6 to 5]

---

> ### Author Response · Authors · 2020-11-17
> **Response to Reviewer 1**
>
> We thank the reviewer for the helpful comments and questions. We respond to the reviewer’s comments one by one.
>
> > Is the Lipschitz constant known? If not, is there a way to test this assumption?
>
> Yes, the Lipschitz constant must be known to the algorithm. Following the reviewer’s suggestion, we have run some experiments where the Lipschitz assumption is violated, i.e., where some congestion functions have Lipschitz constants larger than the Lipschitz constant used by the algorithm. Theoretically, this can result in some bad worst-case scenarios (i.e., linear regret in the worst case), but it seems to improve the experimental performance. Our current hypothesis is that this essentially sacrifices worst-case performance to improve average-case performance.
>
> To elaborate, the reason the algorithm needs to know the Lipschitz constant is in order to make sure that our cost estimates are always underestimates with high probability. This ensures that we don’t accidentally ignore a good edge indefinitely. (The idea is similar to the standard “optimism in the face of uncertainty” principle.) When the Lipschitz assumption is violated (essentially, the algorithm is using too small of a Lipschitz constant), the estimates will not always be underestimates, but will on average be closer to the true value, improving “average case” performance.
>
> It could be interesting to theoretically analyze the “average case” scenario (e.g., if flows are drawn randomly instead of adversarially). We will mention this as an open question in the next revision of the paper.
>
> > I think the authors mean that the regret $R_t$ is $\sum_{r=1}^t E[c_r-c*_r]$.
>
> This is correct, thank you.
>
> > Can you elaborate on your conjecture (e.g., page 3) concerning the right asymptotics of the regret as a function of t? Is your conjecture related to the Awerbuch-Kleinberg SODA paper?
>
> We conjecture that it is impossible to achieve regret $\tilde{O}(\sqrt{n})$ for our problem. The conjecture is related to the Awerbuch-Kleinberg paper in a conceptual sense, but the model is mathematically different, so their results do not directly apply.
>
> To elaborate on the conjecture, reviewer 4 notes that “When f_e is a constant function for each edge, then we recover a version of the stochastic combinatorial semi-bandit.” In that case, $\tilde{O}(\sqrt{n})$ is known to be the best possible, and when estimating the average cost of an arm (i.e., an edge), one can simply use all the samples. However, when $f_e$ is not constant, there is a tension between wanting to use many samples (in order to decrease the effect of noise) and wanting to only use samples y that are close to the target flow x (so that $|f_e(y) - f_e(x)|$ is small). We conjecture that this tension inevitably increases the dependence on T.
>
> > Is the dependence of $|E|$ tight? It felt while reading the proof (e.g., in Lemma A.7 where the summation over time and edges along the path are exchanged) that perhaps the analysis could be improved.
>
> We do not have a tight lower bound for the dependence on $|E|$ and we also conjecture that the linear dependence is not tight (at least not for all graphs). We also do not have a lower bound for the dependence on $\beta$, although we conjecture that the linear dependence on $\beta$ _is_ tight.

---

### Decision · Program_Chairs · 2021-01-07
**Final Decision**

**Decision:**

Reject

**Comment:**

The paper proposes an algorithm with sublinear regret for the problem of routing users through a network with unknown congestion functions over an infinite time horizon. The reviewers generally appreciated the main contribution of this work. One of the reviewers also felt that, although it may be possible to obtain the main result using more standard techniques, it is not clear whether doing so is an easy extension of the prior work. Following the discussion, all of the reviewers agreed that the paper missed important related work and it needs a major revision that incorporates the extensive feedback of Reviewer 2. For these reasons, I recommend reject.